# An Improved Analysis of (Variance-Reduced) Policy Gradient and Natural Policy Gradient Methods

**Yanli Liu**[♮]    **Kaiqing Zhang**[†]    **Tamer Başar**[†]    **Wotao Yin**[♮]
[♮]Department of Mathematics, University of California, Los Angeles
[†]Department of ECE and CSL, University of Illinois at Urbana-Champaign
{yanli, wotaoyin}@math.ucla.edu, {kzhang66, basar1}@illinois.edu

## Abstract

In this paper, we revisit and improve the convergence of policy gradient (PG), natural PG (NPG) methods, and their variance-reduced variants, under general smooth policy parametrizations. More specifically, with the Fisher information matrix of the policy being positive definite: i) we show that a state-of-the-art variance-reduced PG method, which has only been shown to converge to stationary points, converges to the globally optimal value up to some inherent function approximation error due to policy parametrization; ii) we show that NPG enjoys a lower sample complexity; iii) we propose SRVR-NPG, which incorporates variance-reduction into the NPG update. Our improvements follow from an observation that the convergence of (variance-reduced) PG and NPG methods can improve each other: the stationary convergence analysis of PG can be applied to NPG as well, and the global convergence analysis of NPG can help to establish the global convergence of (variance-reduced) PG methods. Our analysis carefully integrates the advantages of these two lines of works. Thanks to this improvement, we have also made variance-reduction for NPG possible, with both global convergence and an efficient finite-sample complexity.

## 1 Introduction

Policy gradient (PG) methods, or more generally direct policy search methods, have long been recognized as one of the foundations of reinforcement learning (RL) [1]. Specifically, PG methods directly search for the optimal policy parameter that maximizes the long-term return in Markov decision processes (MDPs), following the policy gradient ascent direction [2, 3]. This search direction can be more efficient using a preconditioning matrix, e.g., using the natural PG direction [4]. These methods have achieved tremendous empirical successes recently, especially boosted by the power of (deep) neural networks for policy parametrization [5, 6, 7, 8]. These successes are primarily attributed to the fact that PG methods naturally incorporate *function approximation* for policy parametrization, in order to handle massive and even continuous state-action spaces.

In practice, the policy gradients are usually estimated via samples using Monte-Carlo rollouts and bootstrapping [2, 9]. Such stochastic PG methods notoriously suffer from very high variances, which not only destabilize but also slow down the convergence. Several conventional approaches have been advocated to reduce the variance of PG methods, e.g., by adding a baseline [3, 10], or by using function approximation for estimating the value function, namely, developing actor-critic algorithms [11, 12, 13]. More recently, motivated by the advances of variance-reduction techniques in stochastic optimization [14, 15, 16, 17], there have been surging interests in developing *variance-reduced* PG methods [18, 19, 20, 21, 22], which are shown to be faster.

In contrast to the empirical successes of PG methods, their theoretical convergence guarantees, especially *non-asymptotic global* convergence guarantees, have not been addressed satisfactorily

until very recently [23, 24, 25, 26, 27]. By *non-asymptotic global* convergence, here we mean the convergence behavior of PG methods from any initialization, and the quality of the point they converge to (usually enjoys global optimality up to some compatible function approximation error due to policy parametrization), after a finite number of iterations/samples. These recent prominent guarantees are normally beyond the folklore *first-order* stationary-point convergence[1], as expected from a *stochastic nonconvex optimization* perspective of solving RL with PG methods. Special landscapes of the RL objective, though nonconvex, have enabled the convergence to even globally optimal values. On the other hand, none of the aforementioned variance-reduced PG methods [18, 19, 20, 21, 22] have been shown to enjoy these desired global convergence properties. It remains unclear whether these methods can converge to beyond first-order stationary policies.

Motivated by these advances and the questions that remain to be answered, we aim in this paper to improve the convergence of PG and natural PG (NPG) methods, and their variance-reduced variants, under general smooth policy parametrizations. Our contributions are summarized as follows.

**Contributions.** With a focus on the conventional Monte-Carlo-based PG methods, we propose a general framework for analyzing their *global convergence*. Our contribution is three-fold: first, we establish the global convergence up to compatible function approximation errors due to policy parametrization, for a variance-reduced PG method SRVR-PG [21]; second, we improve the global convergence of NPG methods established in [27], from $\mathcal{O}\left(\varepsilon^{-4}\right)$ to $\mathcal{O}\left(\varepsilon^{-3}\right)$; third, we propose a new variance-reduced algorithm based on NPG, and establish its global convergence with an efficient sample-complexity. These improvements are based on a framework that integrates the advantages of previous analyses on (variance reduced) PG and NPG, and rely on a (mild) assumption that the Fisher information matrix induced by the policy parametrization is positive definite (see Assumption 2.1). A comparison of previous results and our improvements is laid out in Table 1.

**Related Work.**

**Global Convergence of (Natural) PG.** Recently, there has been a surging research interest in investigating the global convergence of PG and NPG methods, which is beyond the folklore convergence to first-order stationary policies. In the special case with linear dynamics and quadratic reward, [23] shows that PG methods with random search converge to the globally optimal policy with linear rates. In [24], with a simple reward-reshaping, PG methods have been shown to converge to the second-order stationary-point policies. [26] shows that for finite-MDPs and several control tasks, the nonconvex RL objective has no suboptimal local minima. [25] prove that (natural) PG methods converge to the globally optimal value when overparametrized neural networks are used for function approximation. [27] provides a fairly general characterization of global convergence for these methods, and a basic sample complexity result for sample-based NPG updates. It is also worth noting that trust-region policy optimization (TRPO) [5], as a variant of NPG, also enjoys global convergence with overparametrized neural networks [28], and for regularized MDPs [29]. Very recently, for actor-critic algorithms, a series of non-asymptotic convergence results have also been established [30, 31, 32, 33], with global convergence guarantees when natural PG/PPO are used in the actor step.

**Variance-Reduction (VR) for PG.** Conventional approaches to reduce the high variance in PG methods include using (natural) actor-critic algorithms [11, 12, 13], and adding baselines [3, 10]. The idea of variance reduction (VR) is first proposed to accelerate stochastic minimization. VR algorithms such as SVRG [14, 15, 16], SAGA [17], SARAH [34], and Spider [35] achieve acceleration over SGD in both convex and nonconvex settings. SVRG is also accelerated by applying a positive definite preconditioner that captures the curvature of the objective [36]. Inspired by these successes in stochastic optimization, VR is also incorporated into PG methods [18], with empirical validations for acceleration, and analyzed rigorously in [19]. Then, [20] improves the sample complexity of SVRPG, and [21] proposes a new SRVR-PG method that uses recursively updated semi-stochastic policy gradient, which leads to an improved sample complexity of $\mathcal{O}(\varepsilon^{-1.5})$ over previous works. More recently, [22] proposes a new STORM-PG method, which blends momentum in the update and matches the sample complexity of in [21], and [37] applies the idea of SARAH and considers a more general setting with regularization. Finally, heavy-ball type of momentum has also been applied to PG methods [38]. We highlight that all these sample complexity results are for first-order stationary-point convergence (which might have arbitrarily bad performance: see (2.2)), in contrast

to the more desired global convergence guarantees (up to some function approximation errors that can be small) that we are interested in.

| NPG [27] | NPG [25] | TRPO [28] | TRPO [29] |
|---|---|---|---|
| $\mathcal{O}(\varepsilon^{-4})$ | $\mathcal{O}(T_{TD}\varepsilon^{-2})$ [1] | $\mathcal{O}(\varepsilon^{-8})$ | $\mathcal{O}(\varepsilon^{-4})$ |

| NPG (2.8) | PG (2.4) | SRVR-PG [21] (Algorithm 2) | SRVR-NPG (Algorithm 1) |
|---|---|---|---|
| $\mathcal{O}(\varepsilon^{-3})$ | $\mathcal{O}(\sigma^2\varepsilon^{-4})$ | $\mathcal{O}\big((W + \sigma^2)\varepsilon^{-3}\big)$ | $\mathcal{O}\big((W + \sigma^2)\varepsilon^{-2.5} + \varepsilon^{-3}\big)$ |

Table 1: Comparison of sample complexities of several methods to reach global optimality up to some compatible function approximation error (see (2.9)). Our results are listed in the second table (See App. A for their derivations). We compare the number of trajectories to reach $\varepsilon-$optimality in expectation, up to some inherent error due to the function approximation for policy parametrization (see (2.3)). $\sigma^2$ is an upper bound for the variance of gradient estimator (see Assumption 4.1), and $W$ is an upper bound for the variance of importance weight (see Assumption 4.3).

## 2 Preliminaries

We first introduce some preliminaries regarding both the MDPs and policy gradient methods.

### 2.1 Markov Decision Processes

Consider a discounted Markov decision process defined by a tuple $(\mathcal{S}, \mathcal{A}, \mathbb{P}, R, \gamma)$, where $\mathcal{S}$ and $\mathcal{A}$ denote the state and action spaces of the agent, $\mathbb{P}(s' \,|\, s, a) : \mathcal{S} \times \mathcal{A} \to \mathcal{P}(\mathcal{S})$ is the Markov kernel that determines the transition probability from $(s, a)$ to state $s'$, $\gamma \in (0, 1)$ is the discount factor, and $r : \mathcal{S} \times \mathcal{A} \to [-R, R]$ is the reward function of $s$ and $a$.

At each time $t$, the agent executes an action $a_t \in \mathcal{A}$ given the current state $s_t \in \mathcal{S}$, following a possibly stochastic policy $\pi : \mathcal{S} \to \mathcal{P}(\mathcal{A})$, i.e., $a_t \sim \pi(\cdot \,|\, s_t)$. Then, given the state-action pair $(s_t, a_t)$, the agent observes a reward $r_t = r(s_t, a_t)$. Thus, under any policy $\pi$, one can define the *state-action value* function $Q^\pi : \mathcal{S} \times \mathcal{A} \to \mathbb{R}$ as

$$Q^\pi(s, a) := \mathbb{E}_{a_t \sim \pi(\cdot \,|\, s_t), s_{t+1} \sim \mathbb{P}(\cdot \,|\, s_t, a_t)}\bigg( \sum_{t=0}^\infty \gamma^t r_t \,\bigg|\, s_0 = s, a_0 = a \bigg).$$

One can also define the *state-value* function $V^\pi : \mathcal{S} \to \mathbb{R}$, and the *advantage* function $A^\pi : \mathcal{S} \times \mathcal{A} \to \mathbb{R}$, under policy $\pi$, as $V^\pi(s) := \mathbb{E}_{a \sim \pi(\cdot \,|\, s)}[Q^\pi(s, a)]$ and $A^\pi(s, a) := Q^\pi(s, a) - V^\pi(s)$, respectively. Suppose that the initial state $s_0$ is drawn from some distribution $\rho$. Then, the goal of the agent is to find the optimal policy that maximizes the expected discounted return, namely,

$$\max_\pi \; J(\pi) := \mathbb{E}_{s_0 \sim \rho}[V^\pi(s_0)]. \tag{2.1}$$

In practice, both the state and action spaces $\mathcal{S}$ and $\mathcal{A}$ can be very large. Thus, the policy $\pi$ is usually parametrized as $\pi_\theta$ for some parameter $\theta \in \mathbb{R}^d$, using, for example, deep neural networks. As such, the goal of the agent is to maximize $J(\pi_\theta)$ in the space of the parameter $\theta$, which naturally induces an optimization problem. Such a problem is in general nonconvex [24, 27], making it challenging to find the globally optimal policy.

For notational convenience, let us denote $J(\pi_\theta)$ by $J(\theta)$. Many of the previous works focus on establishing stationary convergence of policy gradient methods. That is, finding a $\theta$ that satisfies

$$\|\nabla J(\theta)\|^2 \le \varepsilon. \tag{2.2}$$

Obviously, such a $\theta$ may not lead to a large $J(\theta)$. Instead, we are interested in finding a $\theta$ such that

$$J^\star - J(\theta) \le \mathcal{O}(\sqrt{\varepsilon_{\text{bias}}}) + \varepsilon, \tag{2.3}$$

where $J^\star = \max_\pi J(\pi)$, and the $\mathcal{O}(\sqrt{\varepsilon_{\text{bias}}})$ term reflects the inherent error related to the possibly limited expressive power of the policy parametrization $\pi_\theta$ (see Assumption 4.4 for the definition).

## 2.2 (Natural) Policy Gradient Methods

To solve the optimization problem (2.1), one standard way is via the policy gradient (PG) method [3]. Specifically, let $\tau_i = \{s_0^i, a_0^i, s_1^i, \cdots\}$ denote the data of a sampled trajectory under policy $\pi_\theta$. Then, a stochastic PG ascent update is given as

$$\theta^{k+1} = \theta^k + \eta \cdot \frac{1}{N} \sum_{i=1}^{N} g(\tau_i \,|\, \theta^k), \tag{2.4}$$

where $\eta > 0$ is a stepsize, $N$ is the number of trajectories, and $g(\tau_i \,|\, \theta^k)$ estimates $\nabla J(\theta^k)$ using the trajectory $\tau_i$. Common unbiased estimators of PG include REINFORCE [2], using the policy gradient theorem [39], and GPOMDP [9]. The commonly used GPOMDP estimator will be given by

$$g(\tau_i \,|\, \theta) = \sum_{h=0}^{\infty} \left( \sum_{t=0}^{h} \nabla_\theta \log \pi_\theta(a_t^i \,|\, s_t^i) \right) \left( \gamma^h r(s_h^i, a_h^i) \right), \tag{2.5}$$

where $\nabla_\theta \log \pi_\theta(a_t^i \,|\, s_t^i)$ is the *score function*. If the expectation of this infinite sum exits, then (2.5) becomes an unbiased estimate of the policy gradient of the objective $J(\theta)$ defined in (2.1). This unbiasedness is established in App. B for completeness.

In practice, a *truncated* version of GPOMDP is used to approximate the infinite sum in (2.5), as

$$g(\tau_i^H \,|\, \theta) = \sum_{h=0}^{H-1} \left( \sum_{t=0}^{h} \nabla_\theta \log \pi_\theta(a_t^i \,|\, s_t^i) \right) \left( \gamma^h r(s_h^i, a_h^i) \right), \tag{2.6}$$

where $\tau_i^H = \{s_0^i, a_0^i, s_1^i, \cdots, s_{H-1}^i, a_{H-1}^i, s_H^i\}$ is a truncation of the full trajectory $\tau_i$ of length $H$. (2.6) is thus a biased stochastic estimate of $\nabla J(\theta)$, with the bias being negligible for a large enough $H$. For notational simplicity, we denote the $H$-horizon trajectory distribution induced by the initial state distribution $\rho$ and policy $\pi_\theta$ as $p_\rho^H(\cdot \,|\, \theta)$, that is,

$$p_\rho^H(\tau^H \,|\, \theta) = \rho(s_0) \prod_{h=0}^{H-1} \pi_\theta(a_h \,|\, s_h) \mathbb{P}(s_{h+1} \,|\, a_h, s_h).$$

Hereafter, unless otherwise stated, we refer to this $H$-*horizon trajectory* simply as *trajectory*, drawn from $p_\rho^H(\cdot \,|\, \theta)$.

As a significant variant of PG, NPG [4] also incorporates a preconditioning matrix $F_\rho(\theta)$, leading to the following update

$$F_\rho(\theta) = \mathbb{E}_{s \sim d_\rho^{\pi_\theta}}[F_s(\theta)], \qquad \theta^{k+1} = \theta^k + \eta \cdot F_\rho^\dagger(\theta^k) \nabla J(\theta^k), \tag{2.7}$$

where $F_s(\theta) = \mathbb{E}_{a \sim \pi_\theta(\cdot \,|\, s)} \left[ \nabla_\theta \log \pi_\theta(a \,|\, s) \nabla_\theta \log \pi_\theta(a \,|\, s)^\top \right]$ is the Fisher information matrix of $\pi_\theta(\cdot \,|\, s) \in \mathcal{P}(\mathcal{A})$, $F_\rho^\dagger(\theta^k)$ is the Moore-Penrose pseudoinverse of $F_\rho(\theta^k)$, and $d_\rho^{\pi_\theta} \in \mathcal{P}(\mathcal{S})$ is the state visitation measure induced by policy $\pi_\theta$ and initial distribution $\rho$, which is defined as

$$d_\rho^{\pi_\theta}(s) := (1 - \gamma) \mathbb{E}_{s_0 \sim \rho} \sum_{t=0}^{\infty} \gamma^t \mathbb{P}(s_t = s \,|\, s_0, \pi_\theta).$$

The NPG update (2.7) can also be written as [4, 27]

$$\theta^{k+1} = \theta^k + \eta \cdot w^k, \quad \text{with} \quad w^k \in \underset{w \in \mathbb{R}^d}{\operatorname{argmin}} \ L_{\nu_\rho^{\pi_\theta}}(w; \theta), \tag{2.8}$$

where $L_{\nu_\rho^{\pi_\theta}}(w; \theta)$ is the *compatible function approximation error* defined by

$$L_{\nu_\rho^{\pi_\theta}}(w; \theta) = \mathbb{E}_{(s,a) \sim \nu_\rho^{\pi_\theta}} \left[ \left( A^{\pi_\theta}(s, a) - (1 - \gamma) w^\top \nabla_\theta \log \pi_\theta(a \,|\, s) \right)^2 \right]. \tag{2.9}$$

Here, $\nu_\rho^{\pi_\theta}(s, a) = d_\rho^{\pi_\theta}(s) \pi(a \,|\, s)$ is the *state-action* visitation measure induced by $\pi_\theta$ and initial state distribution $\rho$, which can also be written as

$$\nu_\rho^{\pi_\theta}(s, a) := (1 - \gamma) \mathbb{E}_{s_0 \sim \rho} \sum_{t=0}^{\infty} \gamma^t \mathbb{P}(s_t = s, a_t = a \,|\, s_0, \pi_\theta). \tag{2.10}$$

For convenience, we will denote $\nu_\rho^{\pi_\theta}$ by $\nu^{\pi_\theta}$ hereafter. In other words, the NPG update direction $w^k$ is given by the minimizer of a stochastic optimization problem. In practice, one obtains an approximate NPG update direction $w^k$ by SGD (see Procedure 1).

Regarding the NPG update (2.8), we make the following standing assumption on the Fisher information matrix induced by $\pi_\theta$ and $\rho$.

**Assumption 2.1.** For all $\theta \in \mathbb{R}^d$, the Fisher information matrix induced by policy $\pi_\theta$ and initial state distribution $\rho$ satisfies

$$F_\rho(\theta) = \mathbb{E}_{(s,a)\sim\nu_\rho^{\pi_\theta}} \left[ \nabla_\theta \log \pi_\theta(a \,|\, s) \nabla_\theta \log \pi_\theta(a \,|\, s)^\top \right] \succcurlyeq \mu_F \cdot I_d$$

for some constant $\mu_F > 0$.

Assumption 2.1 essentially states that $F_\rho(\theta)$ behaves well as a preconditioner in the NPG update (2.8). This is a common (and minimal) requirement for the convergence of preconditioned algorithms in both convex and nonconvex settings in the optimization realm, for example, the quasi-Newton algorithms [40, 41, 42, 43], and their stochastic variants [44, 45, 46, 47, 36]. In the RL realm, one common example of policy parametrizations that can satisfy this assumption is the Gaussian policy [2, 48, 19, 21], where $\pi_\theta(\cdot \,|\, s) = \mathcal{N}(\mu_\theta(s), \Sigma)$ with mean parametrized linearly as $\mu_\theta(s) = \phi(s)^\top \theta$, where $\phi(s)$ denotes some feature matrix of proper dimensions, $\theta$ is the coefficient vector, and $\Sigma \succ 0$ is some fixed covariance matrix. In this case, the Fisher information matrix at each $s$ becomes $\phi(s)\Sigma^{-1}\phi(s)^\top$, independent of $\theta$, and is uniformly lower bounded (positive definite sense) if $\phi(s)$ is full-row-rank, namely, the features expanded by $\theta$ are linearly independent, which is a common requirement for linear function approximation settings [49, 50, 51]. See App. B.2 for more detailed justifications, as well as discussions on more general policy parametrizations.

In the pioneering NPG work [4], $F(\theta)$ is directly assumed to be positive definite. So is in the follow-up works on natural actor-critic algorithms [12, 13]. In fact, this way, $F(\theta)$ will define a valid Riemannian metric on the parameter space, which has been used for interpreting the desired convergence properties of natural gradient methods [52, 53]. In a recent version of [27], a relevant assumption (specifically, Assumption 6.5, item 3) is made to establish the global convergence of NPG, in which it is assumed that $\lambda_{\min}(F_\rho(\theta))$ is not too small compared with the Fisher information matrix induced by a fixed comparator policy. this can be implied by our Assumption 2.1. To sum up, the positive definiteness on the Fisher preconditioning matrix is common and not very restrictive.

In Sec. 4, we shall see that under Assumption 2.1, the stationary convergence of NPG can be analyzed, and NPG enjoys a better sample complexity of $\mathcal{O}(\varepsilon^{-3})$ in terms of its global convergence, compared with the existing sample complexity of $\mathcal{O}(\varepsilon^{-4})$ in [27]. In addition, interestingly, PG and its variance-reduced version SRVR-PG also enjoy global convergence, although the Fisher information matrix does not appear explicitly in their updates.

## 3 Variance-Reduced Policy Gradient Methods

Recently, [21] proposes an algorithm called Stochastic Recursive Variance Reduced Policy Gradient (SRVR-PG, see Algorithm 2), which applies variance-reduction on PG. It achieves a sample complexity of $\mathcal{O}(\varepsilon^{-1.5})$ to find an $\varepsilon-$stationary point, compared with the $\mathcal{O}(\varepsilon^{-2})$ sample complexity of stochastic PG. However, it remains unclear whether SRVR-PG converges globally. In this work, we provide an affirmative answer to this question by showing that SRVR-PG has a sample complexity of $\mathcal{O}(\varepsilon^{-3})$ to find an $\varepsilon-$optimal policy, up to some compatible function approximation error due to policy parametrization.

We also propose a new algorithm called SRVR-NPG to incorporate variance reduction into NPG, which is described in Algorithm 1. In Sec. 4, we provide a sample complexity for its global convergence, which is comparable to our improved NPG result.

In line 8 of Algorithm 1, $g_w(\tau_j^H | \theta_{t-1}^{j+1})$ is a weighted gradient estimator given by

$$g_w(\tau_j^H \,|\, \theta_{t-1}^{j+1}) = \sum_{h=0}^{H-1} w_{0:h}(\tau_j^H \,|\, \theta_{t-1}^{j+1}, \theta_t^{j+1}) \left( \sum_{t=0}^{h} \nabla_\theta \log \pi_\theta(a_t^i \,|\, s_t^i) \right) \left( \gamma^h r(s_h^i, a_h^i) \right), \quad (3.1)$$

**Algorithm 1** Stochastic Recursive Variance Reduced Natural Policy Gradient (SRVR-NPG)

**Input:** number of epochs $S$, epoch size $m$, stepsize $\eta$, batch size $N$, minibatch size $B$, truncation horizon $H$, initial parameter $\theta_m^0 = \theta_0 \in \mathbb{R}^d$ .

1: **for** $j \leftarrow 0, ..., S-1$ **do**
2: $\quad\quad \theta_0^{j+1} = \theta_m^j;$
3: $\quad\quad$ Sample $\{\tau_i^H\}_{i=1}^N$ from $p_\rho^H(\cdot \,|\, \theta_0^{j+1})$ and calculate $u_0^{j+1} = \frac{1}{N}\sum_{i=1}^N g(\tau_i^H \,|\, \theta_0^{j+1});$
4: $\quad\quad w_0^{j+1} = \texttt{SRVR-NPG-SGD}(\nu^{\pi_{\theta_0^{j+1}}}, \pi_{\theta_0^{j+1}}, u_0^{j+1});$ $\quad\quad \triangleright\ w_0^{j+1} \approx w_{0,\star}^{j+1} = F_\rho^{-1}(\theta_0^{j+1})u_0^{j+1};$
5: $\quad\quad \theta_1^{j+1} = \theta_0^{j+1} + \eta w_0^{j+1};$
6: $\quad\quad$ **for** $t \leftarrow 1, ..., m-1$ **do**
7: $\quad\quad\quad\quad$ Sample $B$ trajectories $\{\tau_j^H\}_{j=1}^B$ from $p_\rho^H(\cdot|\theta_t^{j+1});$
8: $\quad\quad\quad\quad u_t^{j+1} = u_{t-1}^{j+1} + \frac{1}{B}\sum_{j=1}^B\left(g(\tau_j^H \,|\, \theta_t^{j+1}) - g_w(\tau_j^H \,|\, \theta_{t-1}^{j+1})\right);$
9: $\quad\quad\quad\quad w_t^{j+1} = \texttt{SRVR-NPG-SGD}(\nu^{\pi_{\theta_t^{j+1}}}, \pi_{\theta_t^{j+1}}, u_t^{j+1});$ $\quad \triangleright\ w_t^{j+1} \approx w_{t,\star}^{j+1} = F_\rho^{-1}(\theta_t^{j+1})u_t^{j+1};$
10: $\quad\quad\quad\quad \theta_{t+1}^{j+1} = \theta_t^{j+1} + \eta w_t^{j+1};$
11: $\quad\quad$ **end for**
12: **end for**
13: **return** $\theta_{\text{out}}$ chosen uniformly from $\{\theta\}_{j=1,...,S;t=0,...,m-1}.$

where the importance weight factor $w_{0:h}(\tau_j^H|\theta_{t-1}^{j+1}, \theta_t^{j+1})$ is defined by

$$w_{0:h}(\tau_j^H \,|\, \theta_{t-1}^{j+1}, \theta_t^{j+1}) = \prod_{h'=0}^h \frac{\pi_{\theta_{t-1}^{j+1}}(a_{h'} \,|\, s_{h'})}{\pi_{\theta_t^{j+1}}(a_{h'} \,|\, s_{h'})}. \tag{3.2}$$

This importance sampling makes $u_t^{j+1}$ an unbiased estimator of $\nabla J^H(\theta_t^{j+1})$.

In lines 4 and 8 of Algorithm 1, $w_t^{j+1}$ is produced by $\texttt{SRVR-NPG-SGD}$ (see Procedure 2), which applies SGD[1] to solve the following subproblem:

$$w_t^{j+1} \approx \underset{w}{\arg\min}\left\{\mathbb{E}_{(s,a)\sim\nu_t^{j+1}}\left[\left(w^T\nabla_\theta\log\pi_{\theta_t^{j+1}}(a\,|\,s)\right)^2\right] - 2\langle w, u_t^{j+1}\rangle\right\}, \tag{3.3}$$

where $\nu_t^{j+1}$ is the state-action visitation measure induced by $\pi_{\theta_t^{j+1}}$. The exact update direction given by (3.3) is $F_\rho^{-1}(\theta_t^{j+1})u_t^{j+1}$, and as in NPG, $F_\rho(\theta_t^{j+1})$ also serves as a preconditioner.

## 4 Theoretical Results

Before presenting the global convergence results, we first introduce some standard assumptions.

**Assumption 4.1.** The truncated GPOMDP estimator $g(\tau^H \,|\, \theta)$ defined in (2.6) satisfies $\text{Var}\left(g(\tau^H \,|\, \theta)\right) := \mathbb{E}[\|g(\tau^H \,|\, \theta) - \mathbb{E}[g(\tau^H \,|\, \theta)]\|^2] \leq \sigma^2$ for any $\theta$ and $\tau^H \sim p_\rho^H(\cdot \,|\, \theta)$.

**Assumption 4.2.** $\quad$ 1. $\|\nabla_\theta\log\pi_\theta(a\,|\,s)\| \leq G$ for any $\theta$ and $(s,a) \in \mathcal{S} \times \mathcal{A}$.

$\quad\quad$ 2. $\|\nabla_\theta\log\pi_{\theta_1}(a\,|\,s) - \nabla_\theta\log\pi_{\theta_2}(a\,|\,s)\| \leq M\|\theta_1 - \theta_2\|$ for any $\theta_1, \theta_2$ and $(s,a) \in \mathcal{S} \times \mathcal{A}$.

**Assumption 4.3.** For the importance weight $w_{0:h}(\tau^H|\theta_1,\theta_2)$ (3.2), there exists $W > 0$ such that

$$\text{Var}(w_{0:h}\left(\tau^H \,|\, \theta_1, \theta_2\right)) \leq W, \quad \forall \theta_1, \theta_2 \in \mathbb{R}^d, \tau^H \sim p_\rho^H(\cdot \,|\, \theta_2).$$

Assumptions 4.1, 4.2 and 4.3 are standard in the analysis of PG methods and their variance reduced variants [27, 19, 20, 21]. They can be verified for simple policy parametrizations such as Gaussian policies; see [19, 57, 58] for more justifications.

Following the Assumption 6.5 of [27], we assume that the policy parametrization $\pi_\theta$ achieves a good function approximation, as measured by the *transferred compatible function approximation error*.

**Assumption 4.4.** For any $\theta \in \mathbb{R}^d$, the *transferred compatible function approximation error* satisfies

$$L_{\nu^\star}(w_\star^\theta; \theta) = \mathbb{E}_{(s,a)\sim\nu^\star}\left[\left(A^{\pi_\theta}(s,a) - (1-\gamma)(w_\star^\theta)^\top \nabla_\theta \log \pi_\theta(a\,|\,s)\right)^2\right] \le \varepsilon_{\text{bias}}, \qquad (4.1)$$

where $\nu^\star(s,a) = d_\rho^{\pi^\star}(s) \cdot \pi^\star(a\,|\,s)$ is the state-action distribution induced by an optimal policy $\pi^\star$ that maximizes $J(\pi)$, and $w_\star^\theta = \operatorname{argmin}_{w\in\mathbb{R}^d} L_{\nu_\rho^{\pi_\theta}}(w;\theta)$ is the exact NPG update direction at $\theta$.

$\varepsilon_{\text{bias}}$ reflects the error when approximating the advantage function from the score function, it measures the capacity of the parametrization $\pi_\theta$. When $\pi_\theta$ is the softmax parametrization, we have $\varepsilon_{\text{bias}} = 0$ [27]. When $\pi_\theta$ is a restricted parametrization, $\varepsilon_{\text{bias}}$ is often positive as $\pi_\theta$ may not contain all stochastic policies. For rich neural parametrizations, $\varepsilon_{\text{bias}}$ is very small [25].

## 4.1 A General Framework for Global Convergence

Inspired by the global convergence analysis of NPG in [27], we present a general framework that relates the global convergence rates of these algorithms to i) their stationary convergence rate on $J(\theta)$, and ii) the difference between their update directions and exact NPG update directions.

**Proposition 4.5.** Let $\{\theta^k\}_{k=1}^K$ be generated by a general update of the form

$$\theta^{k+1} = \theta^k + \eta w^k, \quad k = 0, 1, ... K-1.$$

Furthermore, let $w_\star^k = F_\rho^{-1}(\theta^k)\nabla J(\theta^k)$ be the exact NPG update direction at $\theta^k$. Then, we have

$$J(\pi^\star) - \frac{1}{K}\sum_{k=0}^{K-1} J(\theta^k) \le \frac{\sqrt{\varepsilon_{\text{bias}}}}{1-\gamma} + \frac{1}{\eta K}\mathbb{E}_{s\sim d_\rho^{\pi^\star}}\left[\text{KL}\left(\pi^\star(\cdot\,|\,s)||\pi_{\theta^0}(\cdot\,|\,s)\right)\right]$$

$$+ \frac{M\eta}{2K}\sum_{k=0}^{K-1}\|w^k\|^2 + \frac{G}{K}\sum_{k=0}^{K-1}\|w^k - w_\star^k\|, \qquad (4.2)$$

where $\pi^\star$ is an optimal policy that maximizes $J(\pi)$.

The detailed proof of this global convergence framework can be found in J. To obtain a high level idea, one first starts from the $M-$smoothness of the score function to get

$$\mathbb{E}_{s\sim d_\rho^{\pi^\star}}\left[\text{KL}\left(\pi^\star(\cdot\,|\,s)||\pi_{\theta^k}(\cdot\,|\,s)\right) - \text{KL}\left(\pi^\star(\cdot\,|\,s)||\pi_{\theta^{k+1}}(\cdot\,|\,s)\right)\right]$$

$$\ge \eta\mathbb{E}_{s\sim d_\rho^{\pi^\star}}\mathbb{E}_{a\sim\pi^\star(\cdot\,|\,s)}[\nabla_\theta \log \pi_{\theta^k}(a\,|\,s) \cdot w_\star^k]$$

$$+ \eta\mathbb{E}_{s\sim d_\rho^{\pi^\star}}\mathbb{E}_{a\sim\pi^\star(\cdot\,|\,s)}[\nabla_\theta \log \pi_{\theta^k}(a\,|\,s) \cdot (w^k - w_\star^k)] - \frac{M\eta^2}{2}\|w^k\|^2.$$

On the other hand, the renowned Performance Difference Lemma [59] tells us that

$$\mathbb{E}_{s\sim d_\rho^{\pi^\star}}\mathbb{E}_{a\sim\pi^\star(\cdot\,|\,s)}[A^{\pi_{\theta^k}}(s,a)] = (1-\gamma)\left(J^\star - J(\theta^k)\right).$$

To connect the advantage term $\mathbb{E}_{s\sim d_\rho^{\pi^\star}}\mathbb{E}_{a\sim\pi^\star(\cdot\,|\,s)}[A^{\pi_{\theta^k}}(s,a)]$ with the inner product term $\mathbb{E}_{s\sim d_\rho^{\pi^\star}}\mathbb{E}_{a\sim\pi^\star(\cdot\,|\,s)}[\nabla_\theta \log \pi_{\theta^k}(a\,|\,s) \cdot w_\star^k]$, we invoke Assumption 4.4:

$$\mathbb{E}_{s\sim d_\rho^{\pi^\star}}\mathbb{E}_{a\sim\pi^\star(\cdot\,|\,s)}\left[\left(A^{\pi_\theta}(s,a) - (1-\gamma)(w_\star^\theta)^\top \nabla_\theta \log \pi_\theta(a\,|\,s)\right)^2\right] \le \varepsilon_{\text{bias}}, \quad \text{for any } \theta \in \mathbb{R}^d.$$

The final result follows from a telescoping sum on $k = 0, 1, ..., K-1$.

Several remarks are in order. The first term on the right-hand side of (4.2) reflects the function approximation error due to the parametrization $\pi_\theta$, and the second term is of the form $\mathcal{O}(\frac{1}{K})$. The third term depends on the stationary convergence. With Assumption 2.1, it can be shown that[1] $\frac{1}{K}\sum_{k=0}^{K-1}\mathbb{E}[\|w^k\|^2] \to 0$ for both NPG and SRVR-NPG. The proof follows from an optimization perspective and is inspired by the stationary convergence analysis of stochastic PG (see App. E).

With Assumption 2.1, we can also show that the last term of (4.2) is small. Take stochastic PG as an example; then, we have $w^k = \frac{1}{N}\sum_{i=1}^N g(\tau_i^H|\theta^k)$, and

$$\frac{1}{K}\sum_{k=0}^{K-1}\|w^k - w_\star^k\| \le \frac{1}{K}\sum_{k=0}^{K-1}\|w^k - \nabla J(\theta^k)\| + \frac{1}{K}\sum_{k=0}^{K-1}\left(1 + \frac{1}{\mu_F}\right)\|\nabla J(\theta^k)\|.$$

When $H$ and $N$ are large enough, $w^k$ is a low-variance estimator of $\nabla J^H(\theta^k)$, and $\nabla J^H(\theta^k)$ is close to $\nabla J(\theta^k)$, this makes the first term above small. The second term also goes to 0 as $\theta^k$ approaches stationarity.

## 4.2 Global Convergence Results

By applying Proposition 4.5 on the PG, NPG, SRVR-PG, and SRVR-NPG updates and analyzing their stationary convergence, we obtain their global convergence rates. In the following, we only keep the dependences on $\sigma^2$ (the variance of the gradient estimator), $W$ (variance of importance weight), $\frac{1}{1-\gamma}$ (the effective horizon) and $\varepsilon$ (target accuracy). The specific choice of the parameters and sample complexities, as well as the proof, can be found in the appendix.

**Theorem 4.6.** In the stochastic PG (2.4) with the truncated GPOMDP estimator (2.6), take $\eta = \frac{1}{4L_J}$, $K = \mathcal{O}\left(\frac{1}{(1-\gamma)^2\varepsilon^2}\right)$, $N = \mathcal{O}\left(\frac{\sigma^2}{\varepsilon^2}\right)$, and $H = \mathcal{O}\left(\log(\frac{1}{(1-\gamma)\varepsilon})\right)$. Then, we have

$$J(\pi^\star) - \frac{1}{K}\sum_{k=0}^{K-1}\mathbb{E}[J(\theta^k)] \leq \frac{\sqrt{\varepsilon_{\text{bias}}}}{1-\gamma} + \varepsilon.$$

In total, stochastic PG samples $\mathcal{O}\left(\frac{\sigma^2}{(1-\gamma)^2\varepsilon^4}\right)$ trajectories.

**Remark 4.7.** $L_J = \frac{MR}{(1-\gamma)^2}$ is the Lipschitz constant of $\nabla J$, see Lemma B.1 for details.

**Remark 4.8.** Theorem 4.6 improves the result of [27, Thm. 6.11] from (impractical) full gradients to sample-based stochastic gradients.

**Theorem 4.9.** In the NPG update (2.8), let us apply $\mathcal{O}\left(\frac{1}{(1-\gamma)^4\varepsilon^2}\right)$ iterations of SGD as in Procedure 1 to obtain an update direction. In addition, take $\eta = \frac{\mu_F^2}{4G^2L_J}$ and $K = \mathcal{O}\left(\frac{1}{(1-\gamma)^2\varepsilon}\right)$. Then,

$$J^\star - \frac{1}{K}\sum_{k=0}^{K-1}\mathbb{E}[J(\theta^k)] \leq \frac{\sqrt{\varepsilon_{\text{bias}}}}{1-\gamma} + \varepsilon.$$

In total, NPG samples $\mathcal{O}\left(\frac{1}{(1-\gamma)^6\varepsilon^3}\right)$ trajectories.

**Remark 4.10.** Compared with [27, Coro. 6.10], Theorem 4.9 improves the sample complexity of NPG by $\mathcal{O}(\varepsilon^{-1})$. This is because our stationary convergence analysis on NPG allows for a constant stepsize $\eta$, while [27, Coro. 6.10] applies a stepsize of $\eta = \mathcal{O}(1/\sqrt{K})$. It is worth noting that the $\mathcal{O}(\sqrt{\varepsilon_{\text{bias}}})$ term is the same as in [27], and we also apply the average SGD [54] to solve the NPG subproblem (2.8).

**Theorem 4.11.** In SRVR-PG (Algorithm 2), take $\eta = \frac{1}{8L_J}$, $S = \mathcal{O}\left(\frac{1}{(1-\gamma)^{2.5}\varepsilon}\right)$, $m = \mathcal{O}\left(\frac{(1-\gamma)^{0.5}}{\varepsilon}\right)$, $B = \mathcal{O}\left(\frac{W}{(1-\gamma)^{0.5}\varepsilon}\right)$, $N = \mathcal{O}\left(\frac{\sigma^2}{\varepsilon}\right)$, and $H = \mathcal{O}\left(\log(\frac{1}{(1-\gamma)\varepsilon})\right)$. Then, we have

$$J^\star - \frac{1}{Sm}\sum_{s=0}^{S-1}\sum_{t=0}^{m-1}\mathbb{E}[J(\theta_t^{j+1})] \leq \frac{\sqrt{\varepsilon_{\text{bias}}}}{1-\gamma} + \varepsilon.$$

In total, SRVR-PG samples $\mathcal{O}\left(\frac{W+\sigma^2}{(1-\gamma)^{2.5}\varepsilon^3}\right)$ trajectories.

**Remark 4.12.** Theorem 4.11 establishes the global convergence of SRVR-PG proposed in [21], where only stationary convergence is shown. Also, compared with stochastic PG, SRVR-PG enjoys a better sample complexity thanks to its faster stationary convergence.

**Theorem 4.13.** In SRVR-NPG (Algorithm 1), let us apply $\mathcal{O}\left(\frac{1}{(1-\gamma)^4\varepsilon^2}\right)$ iterations of SGD as in Procedure 2 to obtain an update direction. In addition, take $\eta = \frac{\mu_F}{16L_J}$, $S = \mathcal{O}\left(\frac{1}{(1-\gamma)^{2.5}\varepsilon^{0.5}}\right)$, $m = \mathcal{O}\left(\frac{(1-\gamma)^{0.5}}{\varepsilon^{0.5}}\right)$, $B = \mathcal{O}\left(\frac{W}{(1-\gamma)^{0.5}\varepsilon^{1.5}}\right)$, $N = \mathcal{O}\left(\frac{\sigma^2}{\varepsilon^2}\right)$, and $H = \mathcal{O}\left(\log(\frac{1}{(1-\gamma)\varepsilon})\right)$. Then,

$$J^\star - \frac{1}{Sm}\sum_{s=0}^{S-1}\sum_{t=0}^{m-1}\mathbb{E}[J(\theta_t^{j+1})] \leq \frac{\sqrt{\varepsilon_{\text{bias}}}}{1-\gamma} + \varepsilon.$$

In total, SRVR-NPG samples $\mathcal{O}\left(\frac{W+\sigma^2}{(1-\gamma)^{2.5}\varepsilon^{2.5}} + \frac{1}{(1-\gamma)^6\varepsilon^3}\right)$ trajectories.

**Remark 4.14.** Compared with SRVR-PG, our SRVR-NPG has a better dependence on $W$ and $\sigma^2$, which could be large in practice (especially $W$). The current sample complexity of SRVR-NPG is not better than our (improved) result of NPG since, in our analysis, the advantage of variance reduction is offset by the cost of solving the subproblems.

## 5 Numerical Experiments

In this section, we compare the numerical performances of stochastic PG, NPG, SRVR-PG, and SRVR-NPG. Specifically, we test on benchmark reinforcement learning environments Cartpole and Mountain Car. Our implementation is based on the implementation of SRVPG[1] and SRVR-PG[2], and can be found in the supplementary material.

For both tasks, we apply a Gaussian policy of the form $\pi_\theta(a \mid s) = \frac{1}{\sqrt{2\pi}} \exp\left(-\frac{(\mu_\theta(s)-a)^2}{2\sigma^2}\right)$ where the mean $\mu_\theta(s)$ is modeled by a neural network with Tanh as the activation function.

For the Cartpole problem, we apply a neural network of size $32 \times 1$ and a horizon of $H = 100$. In addition, each training algorithm uses 5000 trajectories in total. For the Mountain Car problem, we apply a neural network of size $64 \times 1$ and take $H = 1000$. 3000 trajectories are allowed for each algorithm. The numerical performance comparison, as well as the settings of algorithm-specific parameters, can be found in Figures 1 and 2. In App. O, we provide more implementation details.

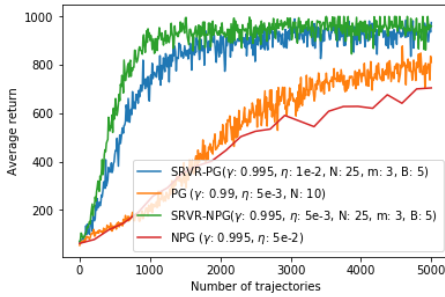
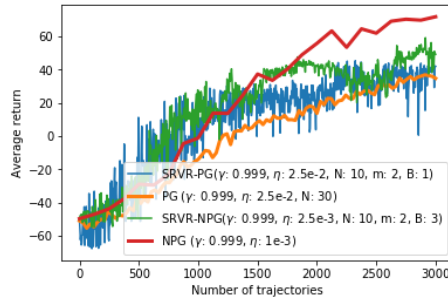

Figure 1: Numerical Performances on Cartpole. For PG, SRVR-PG and SRVR-NPG, we report the undiscounted average return averaged over 10 runs. For NPG, we report the averaged return over 40 runs. Overall, SRVR-NPG has the best performance.

Figure 2: Numerical Performances on Mountain Car. For PG, SRVR-PG and SRVR-NPG, we report the undiscounted average return averaged over 10 runs. For NPG, we report the averaged return over 40 runs. Overall, NPG has the best performance.

## 6 Concluding Remarks

In this work, we have introduced a framework for analyzing the global convergence of (natural) PG methods and their variance-reduced variants, under the assumption that the Fisher information matrix is positive definite. We have established the sample complexity for the global convergence of stochastic PG and its variance-reduced variant SRVR-PG, and improved the sample complexity of NPG. In addition, we have introduced SRVR-NPG, which incorporates variance-reduction into NPG, and enjoys both global convergence guarantee and an efficient sample complexity. Our improved analysis hinges on exploiting the advantages of previous analyses on (variance reduced) PG and NPG methods, which may be of independent interest, and can be used to design faster variance-reduced NPG methods in the future.

## Broader Impact

The results of this paper improves the performance of policy-gradient methods for reinforcement learning, as well as our understanding to the existing methods. Through reinforcement learning, our study will also benefit several research communities such as machine learning and robotics. We do not believe that the results in this work will cause any ethical issue, or put anyone at a disadvantage in our society.

## Acknowledgements

Yanli Liu and Wotao Yin were partially supported by the Office of Naval Research (ONR) Grant N000141712162. Yanli Liu was also supported by UCLA Dissertation Year Fellowship. Kaiqing Zhang and Tamer Başar were supported in part by the US Army Research Laboratory (ARL) Cooperative Agreement W911NF-17-2-0196, and in part by the Office of Naval Research (ONR) MURI Grant N00014-16-1-2710.

## Footnotes

[1]That is, finding a parameter $\theta$ such that $\|\nabla J(\theta)\|^2 \leq \varepsilon$, where $J$ is the expected return.

[1] In [25], $T_{TD}$ iterations of temporal difference updates are needed at each iteration, $T_{TD}$ can be large for wide neural networks. See App. A for details.

[1] Following [27], we apply SGD [54] to make a fair comparison. One can also apply the SA algorithm [55] and AC-SA algorithm [56].

[1]The stationary convergence of SRVR-PG has been established in [21].

[1]`https://github.com/Dam930/rllab`

[2]`https://github.com/xgfelicia/SRVRPG`

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
