[Supplementary Material]

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

 \mid \theta)$ defined in (2.6) satisfies $\operatorname{Var}\big(g(\tau^H \mid \theta)\big) := \mathbb{E}[\|g(\tau^H \mid \theta) - \mathbb{E}[g(\tau^H \mid \theta)]\|^2] \leq \sigma^2$ for any $\theta$ and $\tau^H \sim p_\rho^H(\cdot \mid \theta)$.

**Assumption 4.2.**     1. $\|\nabla_\theta \log \pi_\theta(a \mid s)\| \leq G$ for any $\theta$ and $(s,a) \in \mathcal{S} \times \mathcal{A}$.

2. $\|\nabla_\theta \log \pi_{\theta_1}(a \mid s) - \nabla_\theta \log \pi_{\theta_2}(a \mid s)\| \leq M\|\theta_1 - \theta_2\|$ for any $\theta_1, \theta_2$ and $(s,a) \in \mathcal{S} \times \mathcal{A}$.

**Assumption 4.3.** For the importance weight $w_{0:h}(\tau^H \mid \theta_1, \theta_2)$ (3.2), there exists $W > 0$ such that

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

## A  Derivation of Previous Complexity Bounds

In this section, we briefly explain how to derive the sample complexities bounds in the first line of Table 1.

In the most recent version of [27], a complexity bound of $\mathcal{O}(\varepsilon^{-6})$ can be obtained the taking $N = \mathcal{O}(\varepsilon^{-4})$ and $N = \mathcal{O}(\varepsilon^{-2})$ in its Corollary 6.2. Note this complexity bound can be improved to $\mathcal{O}(\varepsilon^{-4})$ if a uniform upper bound for exact NPG update directions is applied. In this case, one can apply the convergence bound of SGD instead of Projected SGD for the NPG subproblem. In this paper, we establish an upper bound for $\|\nabla J(\theta)\|$ in Lemma B.1. Therefore the exact NPG update direction is also upper bounded thanks to Assumption 2.1.

For [25], the sample complexity bound of $\mathcal{O}(T_{TD}\varepsilon^{-2})$ is achieved by its Theorem 4.13. To be specific, one takes $T = \mathcal{O}(\varepsilon^{-2})$ and $T_{\text{TD}} = \mathcal{O}(m)$ number of temporal difference updates at each iteration. Here, $m$ is width of the neural network.

Note that in the proof of its Corollary 4.14, we can choose $m = \mathcal{O}(T^4)$ (instead of $\mathcal{O}(T^6)$) to have a convergence bound of the form $\mathcal{O}(\sqrt{\varepsilon_0}) + \varepsilon$ (instead of $\mathcal{O}(\varepsilon)$), which is similar to our $\frac{\sqrt{\varepsilon_{\text{bias}}}}{1-\gamma} + \varepsilon$ convergence bound.

For [28], by the Corollary 4.10 therein, one needs to take $K = \mathcal{O}(\varepsilon^{-2})$ and $T = \mathcal{O}(K^3) = \mathcal{O}(\varepsilon^{-6})$, which results in a total sample complexity of $\mathcal{O}(\varepsilon^{-8})$.

For [29], its Theorem 5 (item 1) gives a sample complexity of $\sum_{k=1}^{N} M_k = \mathcal{O}(\varepsilon^{-4})$, where we have applied $N = \mathcal{O}(\varepsilon^{-2})$ and $M_k = \mathcal{O}(\varepsilon^{-2})$.

## B  Helper Lemmas

In this section, we lay out several results that will be useful in later analyses and proofs.

### B.1  Properties of PG Estimator

First, for any $H > 1$, we define the $H$-horizon truncated versions of the return $J(\theta)$ as

$$J^H(\theta) := \mathbb{E}_{s_0 \sim \rho} \left( \sum_{t=0}^{H-1} \gamma^t r_t \right), \tag{B.1}$$

where the expectation is taken over the trajectories, starting from the state distribution $\rho$. Now we establish several properties of the GPOMDP policy gradient estimators and the return functions.

**Lemma B.1.** Recall the GPOMDP policy gradient estimate given in (2.5). The following properties hold:

- If the infinite-sum in (2.5) is well defined, $g(\tau_i \,|\, \theta)$ in (2.5) is an unbiased estimate of the PG $\nabla J(\theta)$. Similarly, the truncated GPOMDP estimate $g(\tau_i^H \,|\, \theta)$ given by (2.6) is an unbiased estimate of the PG $\nabla J^H(\theta)$.

- $J(\theta), J^H(\theta)$ are $L_J$-smooth, where $L_J = \frac{MR}{(1-\gamma)^2}$. Furthermore, we have $\max \left\{ \|\nabla J(\theta)\|, \|\nabla J^H(\theta)\| \right\} \le \frac{GR}{(1-\gamma)^2}$.

- We also have $\|\nabla J^H(\theta) - \nabla J(\theta)\| \le GR \left( \frac{H+1}{1-\gamma} + \frac{\gamma}{(1-\gamma)^2} \right) \gamma^H$.

*Proof.* the unbiasedness of $g(\tau_i \,|\, \theta)$ follows directly from [9]. A similar decomposition can also be done for its truncated version $g(\tau_i^H \,|\, \theta)$.

For the second argument, the smoothness proof is similar to that of Proposition 4.2 in [21]. Specifically, we have know from their Proposition 4.2 that

$$\|\nabla g(\tau_i|\theta)\|_2 \leq \frac{MR}{(1-\gamma)^2}, \qquad \|\nabla g(\tau_i^H|\theta)\|_2 \leq \frac{MR}{(1-\gamma)^2}.$$

Due to the unbiasedness of $g(\tau_i|\theta)$ and $g(\tau_i^H|\theta)$ for estimating $\nabla J(\theta)$ and $\nabla J^H(\theta)$, respectively, by applying $\mathbb{E}[\|\xi\|^2] \geq \|\mathbb{E}[\xi]\|^2$, we know that both $J(\theta)$ and $J^H(\theta)$ are $L_J$-smooth. For obtaining the boundedness of the gradients, similar arguments also apply.

For the third argument, one can calculate that

$$\|g(\tau_i^H \mid \theta) - g(\tau_i \mid \theta)\| = \left\| \sum_{h=H}^{\infty} \left( \sum_{t=0}^{h} \nabla_\theta \log \pi_\theta(a_t^i \mid s_t^i) \right) \left( \gamma^h r(s_h^i, a_h^i) \right) \right\|$$

$$\leq \left\| GR \sum_{h=H}^{\infty} (h+1)\gamma^h \right\|$$

$$= GR \left( \frac{H+1}{1-\gamma} + \frac{\gamma}{(1-\gamma)^2} \right) \gamma^H$$

This rest of the proof follows from the unbiasedness of $g(\tau_i|\theta)$ and $g(\tau_i^H|\theta)$ for estimating $\nabla J(\theta)$ and $\nabla J^H(\theta)$, respectively. $\qquad\square$

## B.2 On the Positive Definiteness of $F_\rho(\theta)$

Now we remark that the positive definiteness on the Fisher information matrix induced by $\pi_\theta$, as stated in Assumption 2.1, is not restricted. Assumption 2.1 essentially states that $F(\theta)$ behaves well as a preconditioner in the NPG update (2.8). This is a common (and minimal) requirement for the convergence of preconditioned algorithms in both convex and nonconvex settings in the optimization realm [44, 45, 46, 47, 36].

In the RL realm, one common example of policy parametrizations that can satisfy this assumption is the Gaussian policy [2, 48, 19, 21], where $\pi_\theta(\cdot \mid s) = \mathcal{N}(\mu_\theta(s), \Sigma)$ with mean parametrized linearly as $\mu_\theta(s) = \phi(s)^\top \theta$, where $\phi(s)$ denotes some feature matrix of proper dimensions, $\theta$ is the coefficient vector, and $\Sigma \succ 0$ is some fixed covariance matrix. Suppose the action $a \in \mathcal{A} \subseteq \mathbb{R}^A$ and recall $\theta \in \mathbb{R}^d$. Thus, $\phi(s) \in \mathbb{R}^{d \times A}$. In this case, the Fisher information at each $s$ becomes $\phi(s)\Sigma^{-1}\phi(s)^\top$, independent of $\theta$, and is positive definite if $\phi(s)$ is full-row-rank. For the case $d < A$, which is usually the case as a lower-dimensional (than $a$) parameter $\theta$ is used, this can be achieved by designing the rows of $\phi(s)$ to be linearly independent, a common requirement for linear function approximation settings [49, 50, 51].

For $\mu_\theta(s)$ being nonlinear functions of $\theta$, e.g., neural networks, the positive definiteness can still be satisfied, if the Jacobian of $\mu_\theta(s)$ at all $\theta$ uniformly satisfies the aforementioned conditions of $\phi(s)$ (the Jacobian in the linear case). In addition, beyond Gaussian policies, with the same conditions mentioned above on the feature $\phi(s)$ or the Jacobian of $\mu_\theta(s)$, Assumption 2.1 also holds more generally for any full-rank exponential family parametrization with mean parametrized by $\mu_\theta(s)$, as the Fisher information matrix, in this case, is also positive definite, in replace of the covariance matrix $\phi(s)\Sigma^{-1}\phi(s)$ in the Gaussian case [60].

Indeed, the Fisher information matrix is positive definite for any *regular* statistical model [61]. In the pioneering NPG work [4], $F(\theta)$ is directly assumed to be positive definite. So is in the follow-up works on natural actor-critic algorithms [12, 13]. In fact, this way, $F_\rho(\theta)$ will define a valid Riemannian metric on the parameter space, which has been used for interpreting the desired convergence properties of natural gradient methods [52, 53]. In sum, the positive definiteness on the Fisher preconditioning matrix is common and not restrictive.

## C SGD and Sampling Procedures

### C.1 SGD for Solving the Subproblems of NPG and SRVR-NPG

Similar to the Algorithm 1 of [27], we also apply the averaged SGD algorithm as in [54] to solve the subproblems of NPG and SRVR-NPG.

---
**Procedure 1** NPG-SGD
---
**Input:** number of iterations $T$, stepsize $\alpha > 0$, objective function $l(w)$, initialization $w_0 = 0$.

  1: **for** $t \leftarrow 0, ..., T-1$ **do**

  2:      $w_{t+1} = w_t - \alpha \widetilde{\nabla} l(w_t)$;          $\triangleright$ $l(w)$ is defined in (C.1), $\widetilde{\nabla} l(w_t)$ is defined in (C.2).

  3: **end for**

  4: **return** $w_{\text{out}} = \frac{1}{T} \sum_{t=1}^{T} w_t$.

---

---
**Procedure 2** SRVR-NPG-SGD
---
**Input:** number of iterations $T$, stepsize $\alpha > 0$, objective function $l(w)$, initialization $w_0 = 0$.

  1: **for** $t \leftarrow 0, ..., T-1$ **do**

  2:      $w_{t+1} = w_t - \alpha \widetilde{\nabla} l(w_t)$;          $\triangleright$ $l(w)$ is defined in (C.3), $\widetilde{\nabla} l(w_t)$ is defined in (C.4).

  3: **end for**

  4: **return** $w_{\text{out}} = \frac{1}{T} \sum_{t=1}^{T} w_t$.

---

For NPG, its subproblem (2.8) is of the form

$$w^k \in \underset{w \in \mathbb{R}^d}{\operatorname{argmin}} \, L_{\nu^{\pi_{\theta^k}}}(w; \theta^k) = \mathbb{E}_{(s,a) \sim \nu^{\pi_{\theta^k}}} \left[ \left( A^{\pi_{\theta^k}}(s,a) - (1-\gamma) w^\top \nabla_\theta \log \pi_{\theta^k}(a \,|\, s) \right)^2 \right],$$

where

$$\nu^{\pi_{\theta^k}}(s,a) = (1-\gamma) \mathbb{E}_{(s_0,a_0) \sim \rho} \sum_{t=0}^{\infty} \gamma^t \mathbb{P}(s_t = s, a_t = a \,|\, s_0, a_0, \pi_{\theta^k}).$$

In Procedure 1, let us set

$$l(w) = \frac{1}{2(1-\gamma)^2} L_{\nu^{\pi_{\theta^k}}}(w; \theta^k). \tag{C.1}$$

Then, we can obtain a stochastic gradient at $w_t$ by

$$\widetilde{\nabla} l(w_t) = \left( (w_t)^T \nabla_\theta \log \pi_{\theta^k}(a|s) - \frac{1}{1-\gamma} \widehat{A}^{\pi_{\theta^k}}(s,a) \right) \nabla_\theta \log \pi_{\theta^k}(a|s) \tag{C.2}$$

where $(s,a) \sim \nu^{\pi_{\theta^k}}$, and $\widehat{A}^{\pi_{\theta^k}}(s,a)$ is an unbiased estimate of $A^{\pi_{\theta^k}}(s,a)$. We will describe how to obtain $(s,a) \sim \nu^{\pi_{\theta^k}}$ and $\widehat{A}^{\pi_{\theta^k}}(s,a)$ in App. C.2.

Following Corollary 6.10 of [27], we can verify that $\widetilde{\nabla} l(w_t)$ is an unbiased estimate of $\nabla l(w_t)$.

For SRVR-NPG, its subproblem (3.3) is of the form

$$w_t^{j+1} \approx \underset{w}{\operatorname{argmin}} \left\{ \mathbb{E}_{(s,a) \sim \nu^{\pi_{\theta_t^{j+1}}}} \left[ \left( w^T \nabla_\theta \log \pi_{\theta_t^{j+1}}(a \,|\, s) \right)^2 \right] - 2 \langle w, u_t^{j+1} \rangle \right\},$$

where

$$\nu^{\pi_{\theta_t^{j+1}}}(s,a) = (1-\gamma) \mathbb{E}_{(s_0,a_0) \sim \rho} \sum_{t=0}^{\infty} \gamma^t \mathbb{P}(s_t = s, a_t = a \,|\, s_0, a_0, \pi_{\theta_t^{j+1}}).$$

In Procedure 2, let us set

$$l(w) = \frac{1}{2} \left( \mathbb{E}_{(s,a) \sim \nu^{\pi_{\theta_t^{j+1}}}} \left[ \left( w^T \nabla_\theta \log \pi_{\theta_t^{j+1}}(a \,|\, s) \right)^2 \right] - 2 \langle w, u_t^{j+1} \rangle \right). \tag{C.3}$$

Then, a stochastic gradient $\widetilde{\nabla} l(w_t)$ is given by

$$\widetilde{\nabla} l(w_t) = \left( (w_t)^T \nabla_\theta \log \pi_{\theta_t^{j+1}}(a|s) \right) \nabla_\theta \log \pi_{\theta_t^{j+1}}(a|s) - u_t^{j+1}. \tag{C.4}$$

where $(s,a) \sim \nu^{\pi_{\theta_t^{j+1}}}$ is obtained in a similar way as above. It is straightforward to verify that $\widetilde{\nabla} l(w_t)$ is an unbiased estimate of $\nabla l(w_t)$.

## C.2 Sampling Procedures

Sampling $(s, a) \sim \nu^{\pi_{\theta^k}}$ and Obtaining $\widehat{A}^{\pi_{\theta^k}}(s, a)$ can be done in a standard way, for example, by apply Algorithm 3 of [27]. Both of them needs to sample $\frac{1}{1-\gamma}$ state-action pairs in expectation.

# D SRVR-PG Algorithm

The Stochastic Recursive Variance-Reduced PG (SRVR-PG) algorithm is introduced in [21], where a recursively updated semi-stochastic gradient $u_t^{j+1}$ is applied as an update direction.

---

**Algorithm 2** Stochastic Recursive Variance Reduced Policy Gradient (SRVR-PG)

---

**Input:** number of epochs $S$, epoch size $m$, stepsize $\eta$, batch size $N$, minibatch size $B$, truncation horizon $H$, initial parameter $\theta_m^0 = \theta_0 \in \mathbb{R}^d$ .

1: **for** $j \leftarrow 0, ..., S-1$ **do**
2:      $\theta_0^{j+1} = \theta_m^j$;
3:      Sample $N$ trajectories $\{\tau_i^H\}_{i=1}^N$ from $p_\rho^H(\cdot|\theta_0^{j+1})$;
4:      $u_0^{j+1} = \frac{1}{N} \sum_{i=1}^N g(\tau_i^H|\theta_0^{j+1})$;
5:      $\theta_1^{j+1} = \theta_0^{j+1} - \eta \nu_0^{j+1}$;
6:      **for** $t \leftarrow 1, ..., m-1$ **do**
7:          Sample $B$ trajectories $\{\tau_j^H\}_{j=1}^B$ from $p_\rho^H(\cdot|\theta_t^{j+1})$;
8:          $u_t^{j+1} = u_{t-1}^{j+1} + \frac{1}{B} \sum_{j=1}^B \left( g(\tau_j^H|\theta_t^{j+1}) - g_w(\tau_j^H|\theta_{t-1}^{j+1}) \right)$;
9:          $\theta_{t+1}^{j+1} = \theta_t^{j+1} - \eta u_t^{j+1}$;
10:      **end for**
11: **end for**
12: **return** $\theta_{\text{out}}$ chosen uniformly from $\{\theta\}_{j=1,...,S;t=0,...,m-1}$.

---

Here, the gradient estimators $g$ and $g_w$ are defined in (2.6) and (3.1), respectively.

# E Stationary Convergence

In this section, we proceed to establish the stationary convergence of stochastic PG, NPG, SRVR-PG, and SRVR-NPG from an optimization perspective.

The stationary convergence of stochastic PG follows from the analysis of SGD. For SRVR-PG, we adapt its analysis in [18].

For NPG and SRVR-NPG, the Fisher information matrix $F(\theta)$ is applied as a preconditioner on top of PG and SRVR-PG, respectively. Regarding $F(\theta)$, we know from Assumptions 2.1 and 4.2 that

$$\mu_F I_d \preccurlyeq F(\theta) \preccurlyeq G^2 I_d \text{ for any } \theta \in \mathbb{R}^d .$$

Since $\mu_F > 0$, we know that $F(\theta)$ defines a nice metric around $\theta$. Consequently, with the analysis of gradient methods in nonconvex optimization, one can show that NPG (SRVR-NPG) has a similar iteration complexity compared with PG (SRVR-PG), although at each iteration, a subproblem needs to be solved in order to obtain an approximate preconditioned update direction.

We next present the stationary convergence results, and prove them in the subsequent sections. These results are established for $J^H(\theta)$ or $J(\theta)$, and we will apply the intermediate results in their proof to establish the global convergence on $J(\theta)$ (up to function approximation errors due to policy parametrizations).

**Theorem E.1.** In the stochastic PG update (2.4), by choosing $\eta = \frac{1}{4L_J}$, $K = \frac{32 L_J (J^{H,\star} - J^H(\theta_0))}{\varepsilon}$, and $N = \frac{6\sigma^2}{\varepsilon}$, we have

$$\frac{1}{K} \sum_{k=0}^{K-1} \mathbb{E}[\|\nabla J^H(\theta^k)\|^2] \leq \varepsilon.$$

In total, stochastic PG samples $\mathcal{O}\left(\frac{\sigma^2}{(1-\gamma)^2\varepsilon^2}\right)$ trajectories.

**Theorem E.2.** In the NPG update (2.8), let us apply $\mathcal{O}\left(\frac{1}{(1-\gamma)^4\varepsilon}\right)$ iterations of SGD as in Procedure 1 to obtain an update direction $w^k$. In addition, let us take $\eta = \frac{\mu_F^2}{4G^2L_J}$ and $K = \frac{32L_JG^4(J^\star-J(\theta_0))}{\mu_F^2\varepsilon}$. Then, we have

$$\frac{1}{K}\sum_{k=0}^{K-1}\mathbb{E}[\|\nabla J(\theta^k)\|^2] \le \varepsilon.$$

In total, NPG samples $\mathcal{O}\left(\frac{1}{(1-\gamma)^6\varepsilon^2}\right)$ trajectories.

**Corollary E.3.** (Theorem 4.5 of [21]) In SRVR-PG (Algorithm 2), take $\eta = \frac{1}{4L_J}$, $N = \frac{12\sigma^2}{\varepsilon}$, $S = \frac{64MR(J^\star-J(\theta^0))}{(1-\gamma)^{2.5}\varepsilon^{0.5}}$, $m = \frac{(1-\gamma)^{0.5}}{\varepsilon^{0.5}}$, and $B = \frac{72\eta G^2(2G^2+M)(W+1)\gamma}{M(1-\gamma)^3}m$. Then, we have

$$\frac{1}{Sm}\sum_{s=0}^{S-1}\sum_{t=0}^{m-1}\mathbb{E}\|\nabla J^H(\theta_t^{j+1})\|^2 \le \varepsilon.$$

In total, SRVR-PG samples $\mathcal{O}\left(\frac{W+\sigma^2}{(1-\gamma)^{2.5}\varepsilon^{1.5}}\right)$ trajectories.

**Theorem E.4.** In SRVR-NPG (Algorithm 1), take $\eta = \frac{\mu_F}{8L_J}$, $S = \frac{24G^2(J^{H,\star}-J^H(\theta_0))}{\eta\varepsilon^{0.5}}$, $m = \frac{1}{\varepsilon^{0.5}}$, $B = \left(\frac{\eta}{\mu_F} + \frac{\eta}{4G^2}\right)\frac{72RG^2(2G^2+M)(W+1)\gamma}{(1-\gamma)^5}\frac{1}{L_J\varepsilon^{0.75}}$, and $N = 3\left(\frac{8G^2}{\mu_F}+2\right)\frac{\sigma^2}{\varepsilon}$. In addition, assume that $\varepsilon$ is small enough such that

$$\varepsilon \le \min\left\{3\left(\frac{8G^2}{\mu_F}+2\right)\left(\frac{GR}{(1-\gamma)^2}\right)^2, 3\left(\frac{8G^2}{4}+\frac{8G^4}{4\mu_F}\right)\frac{2}{\mu_F}\left(\frac{GR}{(1-\gamma)^2}\right)^2,\right.$$
$$\left.\left(\frac{2}{3\eta L_J}(\mu_F+\frac{\mu_F^2}{4G^2})\right)^4\right\}.$$

Let us also apply $\mathcal{O}\left(\frac{1}{(1-\gamma)^4\varepsilon}\right)$ iterations of SGD as in Procedure 2 to obtain an update direction $w_t^{j+1}$. Then, in order to have

$$\frac{1}{Sm}\sum_{s=0}^{S-1}\sum_{t=0}^{m-1}\mathbb{E}\|\nabla J^H(\theta_t^{j+1})\|^2 \le \varepsilon,$$

SRVR-NPG samples $\mathcal{O}\left(\frac{\sigma^2}{(1-\gamma)^2\varepsilon^{1.5}} + \frac{W}{(1-\gamma)^3\varepsilon^{1.75}} + \frac{1}{(1-\gamma)^6\varepsilon^2}\right)$ trajectories.

# F  Proof of Theorem E.1

*Proof.* Let $g^k = \frac{1}{N}\sum_{i=1}^N g(\tau_i^H|\theta^k)$. Then, we have

$$
\begin{aligned}
J^H(\theta^{k+1}) &\ge J^H(\theta^k) + \langle\nabla J^H(\theta^k),\theta^{k+1}-\theta^k\rangle - \frac{L_J}{2}\|\theta^{k+1}-\theta^k\|^2 \\
&= J^H(\theta^k) + \eta\langle\nabla J^H(\theta^k),g^k\rangle - \frac{L_J\eta^2}{2}\|g^k\|^2 \\
&= J^H(\theta^k) + \eta\langle\nabla J^H(\theta^k),g^k-\nabla J^H(\theta^k)+\nabla J^H(\theta^k)\rangle \\
&\quad -\frac{L_J\eta^2}{2}\|g^k-\nabla J^H(\theta^k)+\nabla J^H(\theta^k)\|^2 \\
&\ge J^H(\theta^k) + \frac{\eta}{2}\|\nabla J^H(\theta^k)\|^2 - \frac{\eta}{2}\|g^k-\nabla J^H(\theta^k)\|^2 \\
&\quad -L_J\eta^2\|g^k-\nabla J^H(\theta^k)\|^2 - L_J\eta^2\|\nabla J^H(\theta^k)\|^2 \\
&= J^H(\theta^k) + (\frac{\eta}{2}-L_J\eta^2)\|\nabla J^H(\theta^k)\|^2 - (\frac{\eta}{2}+L_J\eta^2)\|g^k-\nabla J^H(\theta^k)\|^2,
\end{aligned}
\tag{F.1}
$$

where we have applied Lemma B.1 in the first inequality, and Cauchy-Schwartz in the second inequality.

Taking expectation on both sides and applying Lemma B.1 and Assumption 4.1 yields

$$\mathbb{E}[J^H(\theta^{k+1})] \geq \mathbb{E}[J^H(\theta^k)] + (\frac{\eta}{2} - L_J\eta^2)\mathbb{E}[\|\nabla J^H(\theta^k)\|^2] - (\frac{\eta}{2} + L_J\eta^2)\frac{\sigma^2}{N}.$$

Let us further telescope from $k = 0$ to $K - 1$ to obtain

$$\frac{1}{K}\sum_{k=0}^{K-1}\mathbb{E}[\|\nabla J^H(\theta^k)\|^2] \leq \frac{\frac{J^{H,\star}-J^H(\theta_0)}{K} + (\frac{\eta}{2} + L_J\eta^2)\frac{\sigma^2}{N}}{\frac{\eta}{2} - L_J\eta^2}. \tag{F.2}$$

Taking $\eta = \frac{1}{4L_J}$, $K = \frac{32L_J(J^\star - J^H(\theta_0))}{\varepsilon}$, and $N = \frac{6\sigma^2}{\varepsilon}$ gives

$$\frac{1}{K}\sum_{k=0}^{K-1}\mathbb{E}[\|\nabla J^H(\theta^k)\|^2] \leq \varepsilon.$$

Finally, by applying $L_J = \frac{MR}{(1-\gamma)^2}$, we know that PG needs to sample $KN = \frac{192MR(J^{H,\star}-J^H(\theta_0))\sigma^2}{(1-\gamma)^2\varepsilon^2} = \mathcal{O}\left(\frac{\sigma^2}{(1-\gamma)^2\varepsilon^2}\right)$ trajectories. $\qquad\square$

# G   Proof of Theorem E.2

Before proving Theorem E.2, let us first establish the sample complexity of SGD when applied to obtain an approximate NPG update direction $w^k$.

**Proposition G.1.** In Procedure 1, take $\alpha = \frac{1}{4G^2}$ and let the objective be

$$l(w) = \frac{1}{2(1-\gamma)^2}L_{\nu^{\pi_\theta}}(w;\theta^k) = \frac{1}{2}\mathbb{E}_{(s,a)\sim\nu^{\pi_{\theta^k}}}\left[\frac{1}{1-\gamma}A^{\pi_{\theta^k}}(s,a) - w^\top\nabla_\theta\log\pi_{\theta^k}(a\,|\,s)\right]^2.$$

Let $w_\star^k$ be the minimizer of $l(w)$. Then, in order to achieve

$$\mathbb{E}[\|w_{\text{out}} - w_\star^k\|^2] \leq \varepsilon',$$

Procedure 1 requires sampling

$$\frac{4\left(\left(\frac{G^2R}{\mu_F(1-\gamma)^2} + \frac{2}{1-\gamma}\right)\sqrt{d} + \frac{G^2R}{\mu_F(1-\gamma)^2}\right)^2}{\mu_F\varepsilon'} = \mathcal{O}\left(\frac{1}{(1-\gamma)^4\varepsilon'}\right)$$

trajectories.

*Proof.* In this proof, we will suppress the superscript $k$.

Let $l^\star = \min_{w\in\mathbb{R}^d}l(w)$ and $w_\star = \text{argmin}_{w\in\mathbb{R}^d}l(w)$.

By Theorem 1 of [54], we know that

$$\mathbb{E}[l(w_{\text{out}}) - l^\star] \leq \frac{2(\xi\sqrt{d} + G\mathbb{E}[\|w_\star\|])^2}{T},$$

where $l^\star$ is the minimum of $l(w)$, and $\xi$ is defined such that

$$\mathbb{E}[g_\star(g_\star)^T] \preccurlyeq \xi^2\nabla_w^2 l(w),$$

where $g_\star$ is a stochastic gradient of $l(w)$ at $w_\star$.

Following Coro 6.10 of [27], we can take

$$\xi = \frac{G^2R}{\mu_F(1-\gamma)^2} + \frac{2R}{(1-\gamma)^2}.$$

This leads to

$$\mathbb{E}[l(w_{\text{out}}) - l(w_\star)] \leq \frac{2\left(\left(\frac{G^2 R}{\mu_F(1-\gamma)^2} + \frac{2R}{(1-\gamma)^2}\right)\sqrt{d} + \frac{G^2 R}{\mu_F(1-\gamma)^2}\right)^2}{T}.$$

Since $l(w)$ is $\mu_F$−strongly convex, in order to achieve $\mathbb{E}[\|w_{\text{out}} - w_\star^k\|^2] \leq \varepsilon'$, let us set

$$\mathbb{E}[l(w_{\text{out}}) - l(w_\star)] \leq \frac{\mu_F}{2}\varepsilon'.$$

Then, we need

$$T = \frac{4\left(\left(\frac{G^2 R}{\mu_F(1-\gamma)^2} + \frac{2R}{(1-\gamma)^2}\right)\sqrt{d} + \frac{G^2 R}{\mu_F(1-\gamma)^2}\right)^2}{\mu_F \varepsilon'} = \mathcal{O}\left(\frac{1}{(1-\gamma)^4 \varepsilon'}\right).$$

Since each stochastic gradient of SGD has a cost of $\frac{2}{1-\gamma}$ (see App. C), this means to sample $\mathcal{O}\left(\frac{1}{(1-\gamma)^4 \varepsilon'}\right)$ trajectories. $\qquad\qquad\square$

Now, we are ready to prove Theorem E.2.

*Proof of Theorem E.2.* We apply SGD to obtain a $w^k$ such that

$$\mathbb{E}\|w^k - F^{-1}(\theta^k)\nabla J(\theta^k)\|^2 \leq \frac{\mu_F^2 \varepsilon}{32\eta^2 G^4 L_J^2 \left(\frac{2G^4}{\mu_F^2} + 1\right)} = \mathcal{O}\left(\varepsilon\right). \tag{G.1}$$

By Proposition G.1, we need to sample $\mathcal{O}\left(\frac{1}{(1-\gamma)^4 \varepsilon}\right)$ trajectories.

From (G.1) we have

$$\mathbb{E}\|\theta^{k+1} - \theta_\star^{k+1}\|^2 = \eta^2 \mathbb{E}\|w^k - F^{-1}(\theta^k)\nabla J(\theta^k)\|^2 \leq \frac{\mu_F^2 \varepsilon}{32 G^4 L_J^2 \left(\frac{2G^4}{\mu_F^2} + 1\right)}, \tag{G.2}$$

where $\theta_\star^{k+1} = \theta^k + \eta F^{-1}(\theta^k)\nabla J(\theta^k)$.

By Lemma B.1 and Assumption 4.2 we have

$$
\begin{aligned}
J(\theta^{k+1}) &\geq J(\theta^k) + \langle\nabla J(\theta^k), \theta_\star^{k+1} - \theta^k\rangle + \langle\nabla J(\theta^k), \theta^{k+1} - \theta_\star^{k+1}\rangle - \frac{L_J}{2}\|\theta^{k+1} - \theta^k\|^2 \\
&= J(\theta^k) + \eta\langle\nabla J(\theta^k), F^{-1}(\theta^k)\nabla J(\theta^k)\rangle \\
&\quad + \langle\nabla J(\theta^k), \theta^{k+1} - \theta_\star^{k+1}\rangle - \frac{L_J}{2}\|\theta^{k+1} - \theta^k\|^2 \\
&\geq J(\theta^k) + \frac{\eta}{G^2}\|\nabla J(\theta^k)\|^2 + \langle\nabla J(\theta^k), \theta^{k+1} - \theta_\star^{k+1}\rangle - \frac{L_J}{2}\|\theta^{k+1} - \theta^k\|^2.
\end{aligned}
$$

Therefore,

$$
\begin{aligned}
J(\theta^{k+1}) &\geq J(\theta^k) + \frac{\eta}{2G^2}\|\nabla J(\theta^k)\|^2 - \frac{G^2}{2\eta}\|\theta^{k+1} - \theta_\star^{k+1}\|^2 - \frac{L_J}{2}\|\theta^{k+1} - \theta^k\|^2 \\
&\geq J(\theta^k) + \frac{\eta}{2G^2}\|\nabla J(\theta^k)\|^2 - \left(\frac{G^2}{2\eta} + L_J\right)\|\theta^{k+1} - \theta_\star^{k+1}\|^2 - L_J\|\theta_\star^{k+1} - \theta^k\|^2 \\
&\geq J(\theta^k) + \left(\frac{\eta}{2G^2} - \frac{L_J \eta^2}{\mu_F^2}\right)\|\nabla J(\theta^k)\|^2 - \left(\frac{G^2}{2\eta} + L_J\right)\|\theta^{k+1} - \theta_\star^{k+1}\|^2,
\end{aligned}
$$

where we have applied Cauchy-Schwartz in the first and second inequalities, and $\theta_\star^{k+1} = \theta^k + \eta F^{-1}(\theta^k)\nabla J(\theta^k)$ in the last step.

Taking full expectation on both sides yields

$$\mathbb{E}[J(\theta^{k+1})] \geq \mathbb{E}[J(\theta^k)] + \left(\frac{\eta}{2G^2} - \frac{L_J\eta^2}{\mu_F^2}\right)\mathbb{E}\|\nabla J(\theta^k)\|^2 - \left(\frac{G^2}{2\eta} + L_J\right)\mathbb{E}\|\theta^{k+1} - \theta_\star^{k+1}\|^2$$

$$\geq \mathbb{E}[J(\theta^k)] + \left(\frac{\eta}{2G^2} - \frac{L_J\eta^2}{\mu_F^2}\right)\mathbb{E}\|\nabla J(\theta^k)\|^2 - \left(\frac{G^2}{2\eta} + L_J\right)\frac{\mu_F^2\varepsilon}{32G^4L_J^2\left(\frac{2G^4}{\mu_F^2}+1\right)},$$

where we have applied (G.2) in the second inequality.

Telescoping the above inequality from $k = 0$ to $k = K - 1$ gives

$$\frac{J^\star - J(\theta_0)}{K} \geq \left(\frac{\eta}{2G^2} - \frac{L_J\eta^2}{\mu_F^2}\right)\frac{1}{K}\sum_{k=0}^{K-1}\mathbb{E}\|\nabla J(\theta^k)\|^2 - \left(\frac{G^2}{2\eta} + L_J\right)\frac{\mu_F^2\varepsilon}{32G^4L_J^2\left(\frac{2G^4}{\mu_F^2}+1\right)}.$$

Finally, by taking $\eta = \frac{\mu_F^2}{4G^2L_J}$ and $K = \frac{32L_JG^4(J^\star-J(\theta_0))}{\mu_F^2\varepsilon} = \mathcal{O}\left(\frac{1}{(1-\gamma)^2\varepsilon}\right)$, we arrive at

$$\frac{1}{K}\sum_{k=0}^{K-1}\mathbb{E}[\|\nabla J(\theta^k)\|^2] \leq \frac{\frac{J^\star-J(\theta_0)}{K} + \frac{\left(\frac{G^2}{2\eta}+L_J\right)\mu_F^2\varepsilon}{32G^4L_J^2\left(\frac{2G^4}{\mu_F^2}+1\right)}}{\left(\frac{\eta}{2G^2} - \frac{L_J\eta^2}{\mu_F^2}\right)} = \varepsilon. \qquad \text{(G.3)}$$

Recall that at each iteration of NPG, we apply SGD as in Procedure 1 to reach (G.1). By Proposition G.1, we know that in total, NPG requires to sample

$$\frac{32L_JG^4(J^\star - J(\theta_0))}{\mu_F^2\varepsilon} \cdot \frac{4\left(\left(\frac{G^2R}{\mu_F(1-\gamma)^2} + \frac{2R}{(1-\gamma)^2}\right)\sqrt{d} + \frac{G^2R}{\mu_F(1-\gamma)^2}\right)^2}{\mu_F\frac{\mu_F^2\varepsilon}{32\eta^2G^4L_J^2\left(\frac{2G^4}{\mu_F^2}+1\right)}} = \mathcal{O}\left(\frac{1}{(1-\gamma)^6\varepsilon^2}\right)$$

trajectories.

$\square$

# H    Proof of Theorem E.3

*Proof.* By Theorem 4.5 of [21], we know that if $\eta = \frac{1}{4L_J}$ and

$$B = \frac{3\eta C_\gamma m}{L_J} = \frac{72\eta G^2(2G^2 + M)(W+1)\gamma}{M(1-\gamma)^3}m,$$

then

$$\frac{1}{Sm}\sum_{s=0}^{S-1}\sum_{t=0}^{m-1}\mathbb{E}\|\nabla J^H(\theta_t^{j+1})\|^2 \leq \frac{8(J^\star - J^H(\theta_0))}{\eta Sm} + \frac{6\sigma^2}{N}.$$

Therefore, taking $N = \frac{12\sigma^2}{\varepsilon}$ and $Sm = \frac{64MR(J^\star - J^H(\theta^0))}{(1-\gamma)^2\varepsilon}$ yields

$$\frac{1}{Sm}\sum_{s=0}^{S-1}\sum_{t=0}^{m-1}\mathbb{E}\|\nabla J^H(\theta_t^{j+1})\|^2 \leq \varepsilon.$$

Let us take $S = \frac{64MR(J^\star - J^H(\theta^0))}{(1-\gamma)^{2.5}\varepsilon^{0.5}}$ and $m = \frac{(1-\gamma)^{0.5}}{\varepsilon^{0.5}}$. Then, the number of trajectories required by SRVR-PG is

$$S(N + mB) = S\frac{12\sigma^2}{\varepsilon} + \frac{64MR(J^\star - J^H(\theta^0))}{(1-\gamma)^2\varepsilon}B$$

$$= S\frac{12\sigma^2}{\varepsilon} + \frac{64MR(J^\star - J^H(\theta^0))}{(1-\gamma)^2\varepsilon}\frac{72\eta G^2(2G^2+M)(W+1)\gamma}{M(1-\gamma)^3}m$$

$$= \mathcal{O}\left(\frac{\sigma^2}{(1-\gamma)^{2.5}\varepsilon^{1.5}} + \frac{W}{(1-\gamma)^{2.5}\varepsilon^{1.5}}\right)$$

$$= \mathcal{O}\left(\frac{W+\sigma^2}{(1-\gamma)^{2.5}\varepsilon^{1.5}}\right).$$

Therefore, SRVR-PG needs to sample $\mathcal{O}\left(\frac{W+\sigma^2}{(1-\gamma)^{2.5}\varepsilon^{1.5}}\right)$ trajectories. $\qquad\square$

# I    Proof of Theorem E.4

In order to prove Theorem E.4, we need the following technical results.

**Lemma I.1** (Equation B.10 of [20])**.** We have

$$\mathbb{E}\|\nabla J^H(\theta_t^{j+1}) - u_t^{j+1}\|^2 \leq \frac{C_\gamma}{B}\sum_{l=1}^{t}\mathbb{E}\|\theta_l^{j+1} - \theta_{l-1}^{j+1}\|^2 + \frac{\sigma^2}{N},$$

where

$$C_\gamma = \frac{24RG^2(2G^2+M)(W+1)\gamma}{(1-\gamma)^5}.$$

*Proof.* This lemma is adapted from the Equation B.10 of [20], where SRVR-PG is analyzed. It is also true for our SRVR-NPG since the update rule of $u_t^{j+1}$ is the same for both algorithms. $\qquad\square$

**Proposition I.2.** In SRVR-NPG, apply SGD as in Procedure 2 to solve the subproblems. Take $\alpha = \frac{1}{4G^2}$ and let the objective be

$$l(w) = \frac{1}{2}\left(\mathbb{E}_{(s,a)\sim\nu_t^{j+1}}[w^T\nabla_\theta\log\pi_{\theta_t^{j+1}}(a\,|\,s)]^2 - 2\langle\eta w, u_t^{j+1}\rangle\right).$$

Let $w_{t,\star}^{j+1} = F^{-1}(\theta_t^{j+1})\nabla J^H(\theta_t^{j+1})$ be the minimizer of $l(w)$. Assume in addition that

$$\frac{\sigma^2}{N} \leq \left(\frac{GR}{(1-\gamma)^2}\right)^2,$$

$$\varepsilon' \leq \frac{2}{\mu_F^2}\left(\frac{GR}{(1-\gamma)^2}\right)^2$$

$$\frac{C_\gamma m}{B}2\eta^2 \leq \frac{1}{3}\mu_F^2.$$

Then, in order to achieve

$$\mathbb{E}[\|w_t^{j+1} - w_{t,\star}^{j+1})\|^2] \leq \varepsilon'$$

for each $s = 0, 1, ..., S-1$ and $t = 0, 1, ..., m-1$, Procedure 2 requires sampling

$$\frac{4\left(\frac{\frac{2}{\mu_F}\frac{GR}{(1-\gamma)^2}G^2 + \frac{2GR}{(1-\gamma)^2}}{\sqrt{\mu_F}}\sqrt{d} + \frac{2G^2R}{\mu_F(1-\gamma)^2}\right)^2}{\mu_F\varepsilon'} = \mathcal{O}\left(\frac{1}{(1-\gamma)^4\varepsilon'}\right)$$

trajectories.

*Proof of Proposition I.2.* Recall that we are applying SGD as in Procedure 2 to solve the SRVR-NPG subproblem (3.3).

Let us focus on $t = 0$, where $u_0^{j+1} = \frac{1}{N}\sum_{i=1}^{N}g(\tau_i^H|\theta_0^{j+1})$. As a result, $\mathbb{E}[u_0^{j+1}] = \nabla J^H(\theta_0^{j+1})$ and $\text{Var}(u_0^{j+1}) \leq \frac{\sigma^2}{N}$. Therefore,

$$\mathbb{E}[\|w_{0,\star}^{j+1}\|^2] = \mathbb{E}[\|F^{-1}(\theta_0^{j+1})u_0^{j+1}\|^2]$$

$$\leq \frac{1}{\mu_F^2}\mathbb{E}[\|u_0^{j+1}\|] \leq \frac{1}{\mu_F^2}\left(\frac{GR}{(1-\gamma)^2}\right)^2 + \frac{1}{\mu_F^2}\frac{\sigma^2}{N} \leq \frac{4}{\mu_F^2}\left(\frac{GR}{(1-\gamma)^2}\right)^2.$$

Recall from (C.4) that a stochastic gradient $\nabla l(w_t)$ is given by

$$\nabla l(w) = \left(w^T\nabla_\theta\log\pi_{\theta_0^{j+1}}(a|s)\right)\nabla_\theta\log\pi_{\theta_0^{j+1}}(a|s) - u_0^{j+1}.$$

Here, $s, a \sim \nu^{\pi_\theta}$.

Therefore, By Theorem 1 of [54], we know that in order to reach $\mathbb{E}[\|w_0^{j+1} - (w_0^{j+1})_\star\|^2] \leq \varepsilon'$, we need

$$\mathbb{E}[l(w_{\text{out}}) - l^\star] \leq \frac{2(\xi\sqrt{d} + G\|w_{0,\star}^{j+1}\|)^2}{T} \leq \frac{\mu_F}{2}\varepsilon',$$

where $l^\star$ is the minimum of $l(w)$, and $\xi$ is defined such that the stochastic gradient $g_\star$ at the solution $w_{0,\star}^{j+1}$ satisfies

$$\mathbb{E}[g_\star(g_\star)^T] \preccurlyeq \xi^2 \nabla_w^2 l(w).$$

Similar as Proposition G.1, we know that $\xi$ can be chosen by

$$\xi^2 = \frac{\left(\frac{2}{\mu_F}\frac{GR}{(1-\gamma)^2}G^2 + \frac{2GR}{(1-\gamma)^2}\right)^2}{\mu_F}.$$

As a result, the number of iterations, $T$, should be

$$T = \frac{4\left(\frac{\frac{2}{\mu_F}\frac{GR}{(1-\gamma)^2}G^2 + \frac{2GR}{(1-\gamma)^2}}{\sqrt{\mu_F}}\sqrt{d} + \frac{2G^2R}{\mu_F(1-\gamma)^2}\right)^2}{\mu_F\varepsilon'} = \mathcal{O}\left(\frac{1}{(1-\gamma)^4\varepsilon'}\right).$$

Since each stochastic gradient of $l(w)$ only needs to sample a state-action pair, this is equivalent to sampling $\mathcal{O}\left(\frac{1}{(1-\gamma)^3\varepsilon'}\right)$ trajectories.

Now, let us turn to $t \geq 1$. $u_t^{j+1}$ is an unbiased estimate of $\nabla J(\theta_t^{j+1})$, and its variance is bounded as in Lemma I.1. Therefore,

$$
\begin{aligned}
\mathbb{E}[\|w_{t,\star}^{j+1})\|^2] = \mathbb{E}[\|F^{-1}(\theta_t^{j+1})u_t^{j+1}\|^2] &\leq \frac{1}{\mu_F^2}\mathbb{E}[\|u_t^{j+1}\|] \\
&= \frac{1}{\mu_F^2}\mathbb{E}[\|\nabla J^H(\theta_t^{j+1})\|^2] + \frac{1}{\mu_F^2}\mathbb{E}[\|u_t^{j+1} - \nabla J^H(\theta_t^{j+1})\|^2] \\
&\leq \frac{1}{\mu_F^2}\left(\frac{GR}{(1-\gamma)^2}\right)^2 + \frac{1}{\mu_F^2}\left(\frac{C_\gamma}{B}\sum_{l=1}^t \mathbb{E}\|\theta_l^{j+1} - \theta_{l-1}^{j+1}\|^2 + \frac{\sigma^2}{N}\right) \\
&= \frac{1}{\mu_F^2}\left(\frac{GR}{(1-\gamma)^2}\right)^2 + \frac{1}{\mu_F^2}\left(\frac{C_\gamma}{B}\sum_{l=1}^t \eta^2\mathbb{E}\|w_{l-1}^{j+1}\|^2 + \frac{\sigma^2}{N}\right) \\
&\leq \frac{1}{\mu_F^2}\left(\frac{GR}{(1-\gamma)^2}\right)^2 \\
&\quad + \frac{1}{\mu_F^2}\left(\frac{C_\gamma}{B}\sum_{l=1}^t 2\eta^2\left(\mathbb{E}\|w_{l-1,\star}^{j+1}\|^2 + \mathbb{E}\|w_{l-1,\star}^{j+1} - w_{l-1}^{j+1}\|^2\right) + \frac{\sigma^2}{N}\right).
\end{aligned}
$$

(I.1)

Now, we are ready to prove the desired results by induction.

Assume that for all $t' < t$, we have

$$\mathbb{E}[\|w_{t',\star}^{j+1})\|] \leq \frac{4}{\mu_F^2}\left(\frac{GR}{(1-\gamma)^2}\right)^2,$$

and we have applied

$$T = \frac{4\left(\frac{\frac{2}{\mu_F}\frac{GR}{(1-\gamma)^2}G^2 + \frac{2GR}{(1-\gamma)^2}}{\sqrt{\mu_F}}\sqrt{d} + \frac{2G^2R}{\mu_F(1-\gamma)^2}\right)^2}{\mu_F\varepsilon'} = \mathcal{O}\left(\frac{1}{(1-\gamma)^4\varepsilon'}\right)$$

iterations of SGD as in Procedure 2.

Similar to the case of $t = 0$, we know that this yields

$$\mathbb{E}[\|w_{t'}^{j+1} - w_{t',\star}^{j+1}\|^2] \leq \varepsilon'.$$

Then, by (I.1), we have

$$\mathbb{E}[\|w_{t,\star}^{j+1}\|^2] \leq \frac{1}{\mu_F^2}\left(\frac{GR}{(1-\gamma)^2}\right)^2$$

$$+ \frac{1}{\mu_F^2}\left(\frac{C_\gamma}{B}\sum_{l=1}^{t}2\eta^2\left(\mathbb{E}\|w_{l-1,\star}^{j+1}\|^2 + \mathbb{E}\|w_{l-1,\star}^{j+1} - w_{l-1}^{j+1}\|^2\right) + \frac{\sigma^2}{N}\right)$$

$$\leq \frac{1}{\mu_F^2}\left(\frac{GR}{(1-\gamma)^2}\right)^2 + \frac{1}{\mu_F^2}\left(\frac{C_\gamma m}{B}2\eta^2\left(\frac{4}{\mu_F^2}\left(\frac{GR}{(1-\gamma)^2}\right)^2 + \varepsilon'\right) + \frac{\sigma^2}{N}\right)$$

$$= \frac{1}{\mu_F^2}\left(\frac{GR}{(1-\gamma)^2}\right)^2 + \frac{1}{\mu_F^2}\frac{C_\gamma m}{B}2\eta^2\frac{4}{\mu_F^2}\left(\frac{GR}{(1-\gamma)^2}\right)^2$$

$$+ \frac{1}{\mu_F^2}\frac{C_\gamma m}{B}2\eta^2\varepsilon' + \frac{1}{\mu_F^2}\frac{\sigma^2}{N}$$

$$\leq \frac{4}{\mu_F^2}\left(\frac{GR}{(1-\gamma)^2}\right)^2.$$

As a result, we can apply

$$T = \frac{4\left(\frac{\frac{2}{\mu_F}\frac{GR}{(1-\gamma)^2}G^2 + \frac{2GR}{(1-\gamma)^2}}{\sqrt{\mu_F}}\sqrt{d} + \frac{2G^2R}{\mu_F(1-\gamma)^2}\right)^2}{\mu_F\varepsilon'} = \mathcal{O}\left(\frac{1}{(1-\gamma)^4\varepsilon'}\right)$$

iterations of SGD as in Procedure 2 so that

$$\mathbb{E}[\|w_t^{j+1} - w_{t,\star}^{j+1}\|^2] \leq \varepsilon'.$$

Since each stochastic gradient of $l(w)$ has a cost of $\frac{1}{1-\gamma}$ (see App. C), this is equivalent to sample

$$\frac{4\left(\frac{\frac{2}{\mu_F}\frac{GR}{(1-\gamma)^2}G^2 + \frac{2GR}{(1-\gamma)^2}}{\sqrt{\mu_F}}\sqrt{d} + \frac{2G^2R}{\mu_F(1-\gamma)^2}\right)^2}{\mu_F\varepsilon'} = \mathcal{O}\left(\frac{1}{(1-\gamma)^4\varepsilon'}\right)$$

trajectories. $\qquad\square$

We are now ready to prove Theorem E.4.

*Proof of Theorem E.4.* Line 9 of Algorithm 1 reads

$$w_t^{j+1} \approx \underset{w}{\mathrm{argmin}}\{\mathbb{E}_{(s,a)\sim\nu_t^{j+1}}[w^T\nabla_\theta\log\pi_{\theta_t^{j+1}}(s,a)]^2 - 2\langle\eta w, u_t^{j+1}\rangle\}.$$

And we want to apply SGD as in Procedure 2 to obtain a $w_t^{j+1}$ that satisfies

$$\mathbb{E}[\|w_t^{j+1} - F^{-1}(\theta_t^{j+1})u_t^{j+1}\|^2] \leq \frac{\varepsilon}{3\left(\frac{8G^2\mu_F}{4} + \frac{8G^4}{4}\right)}. \tag{I.2}$$

Recall that the parameters $S, m, B$ and $N$ are chosen as

$$S = \frac{24G^2(J^\star - J^H(\theta_0))}{\eta\varepsilon^{0.5}},$$

$$m = \frac{1}{\varepsilon^{0.5}},$$

$$B = \left(\frac{\eta}{\mu_F} + \frac{\eta}{4G^2}\right)\frac{4C_\gamma m}{L_J\varepsilon^{0.25}},$$

$$N = 3\left(\frac{8G^2}{\mu_F} + 2\right)\frac{\sigma^2}{\varepsilon}.$$

Since

$$\varepsilon \leq \min\left\{3\left(\frac{8G^2}{\mu_F} + 2\right)\left(\frac{GR}{(1-\gamma)^2}\right)^2, 3\left(\frac{8G^2}{4} + \frac{8G^4}{4\mu_F}\right)\frac{2}{\mu_F}\left(\frac{GR}{(1-\gamma)^2}\right)^2, \right.$$
$$\left.\left(\frac{2}{3\eta L_J}(\mu_F + \frac{\mu_F^2}{4G^2})\right)^4\right\},$$

the requirements of Proposition I.2 are satisfied:

$$\frac{\sigma^2}{N} = \frac{\varepsilon}{3\left(\frac{8G^2}{\mu_F} + 2\right)} \leq \left(\frac{GR}{(1-\gamma)^2}\right)^2,$$

$$\varepsilon' = \frac{\varepsilon}{3\left(\frac{8G^2\mu_F}{4} + \frac{8G^4}{4}\right)} \leq \frac{2}{\mu_F^2}\left(\frac{GR}{(1-\gamma)^2}\right)^2$$

$$\frac{C_\gamma m}{B}2\eta^2 = \frac{2\eta^2\varepsilon^{0.25}}{\left(\frac{\eta}{\mu_F} + \frac{\eta}{4G^2}\right)\frac{4}{L_J}} \leq \frac{1}{3}\mu_F^2.$$

By applying Proposition I.2, we know that in order to have (I.2), one needs to sample

$$\frac{4\left(\frac{\frac{\sqrt{2}}{\mu_F}\frac{GR}{(1-\gamma)^2}G^2 + \frac{\sqrt{2}GR}{(1-\gamma)^2}}{\sqrt{\mu_F}}\sqrt{d} + \frac{\sqrt{2}G^2R}{\mu_F(1-\gamma)^2}\right)^2}{\mu_F\varepsilon'} = \mathcal{O}\left(\frac{1}{(1-\gamma)^4\varepsilon}\right)$$

trajectories. By (I.2) we know that

$$\mathbb{E}[\|\theta_{t+1}^{j+1} - \theta_{t+1,\star}^{j+1}\|^2] \leq \frac{\varepsilon}{3\left(\frac{8G^2\mu_F}{4\eta^2} + \frac{8G^4}{4\eta^2}\right)} \tag{I.3}$$

where $\theta_{t+1,\star}^{j+1} = \theta_t^{j+1} + \eta F^{-1}(\theta_t^{j+1})u_t^{j+1}$.
On the other hand, we have

$$J^H(\theta_{t+1}^{j+1}) \geq J^H(\theta_t^{j+1}) + \langle \nabla J^H(\theta_t^{j+1}), \theta_{t+1}^{j+1} - \theta_t^{j+1}\rangle - \frac{L_J}{2}\|\theta_{t+1}^{j+1} - \theta_t^{j+1}\|^2$$
$$= J^H(\theta_t^{j+1}) + \langle \nabla J^H(\theta_t^{j+1}) - u_t^{j+1}, \theta_{t+1}^{j+1} - \theta_t^{j+1}\rangle$$
$$+ \langle u_t^{j+1}, \theta_{t+1}^{j+1} - \theta_t^{j+1}\rangle - \frac{L_J}{2}\|\theta_{t+1}^{j+1} - \theta_t^{j+1}\|^2$$
$$\geq J^H(\theta_t^{j+1}) - \frac{\eta}{\mu_F}\|\nabla J^H(\theta_t^{j+1}) - u_t^{j+1}\|^2 - \frac{\mu_F}{4\eta}\|\theta_{t+1}^{j+1} - \theta_t^{j+1}\|^2$$
$$+ \langle u_t^{j+1}, \theta_{t+1}^{j+1} - \theta_t^{j+1}\rangle - \frac{L_J}{2}\|\theta_{t+1}^{j+1} - \theta_t^{j+1}\|^2$$

where we have applied Lemma B.1 in the first inequality, and Cauchy-Schwartz in the second one.

Rearranging gives

$$
\begin{aligned}
J^H(\theta_{t+1}^{j+1}) \geq{}& J^H(\theta_t^{j+1}) - \frac{\eta}{\mu_F}\|\nabla J^H(\theta_t^{j+1}) - u_t^{j+1}\|^2 - \frac{\mu_F}{4\eta}\|\theta_{t+1}^{j+1} - \theta_t^{j+1}\|^2 \\
&+ \frac{1}{2}\langle u_t^{j+1}, \theta_{t+1,\star}^{j+1} - \theta_t^{j+1}\rangle \\
&+ \frac{1}{2}\langle u_t^{j+1}, \theta_{t+1}^{j+1} - \theta_{t+1,\star}^{j+1}\rangle + \frac{1}{2}\langle \frac{1}{\eta}F(\theta_t^{j+1})(\theta_{t+1,\star}^{j+1} - \theta_t^{j+1}), \theta_{t+1}^{j+1} - \theta_t^{j+1}\rangle \\
&- \frac{L_J}{2}\|\theta_{t+1}^{j+1} - \theta_t^{j+1}\|^2 \\
={}& J^H(\theta_t^{j+1}) - \frac{\eta}{\mu_F}\|\nabla J^H(\theta_t^{j+1}) - u_t^{j+1}\|^2 \\
&- \frac{\mu_F}{4\eta}\|\theta_{t+1}^{j+1} - \theta_t^{j+1}\|^2 + \frac{1}{2}\langle u_t^{j+1}, \eta F^{-1}(\theta_t^{j+1})u_t^{j+1}\rangle \\
&+ \frac{1}{2}\langle u_t^{j+1}, \theta_{t+1}^{j+1} - \theta_{t+1,\star}^{j+1}\rangle + \frac{1}{2}\langle \frac{1}{\eta}F(\theta_t^{j+1})(\theta_{t+1,\star}^{j+1} - \theta_t^{j+1}), \theta_{t+1}^{j+1} - \theta_t^{j+1}\rangle \\
&- \frac{L_J}{2}\|\theta_{t+1}^{j+1} - \theta_t^{j+1}\|^2,
\end{aligned}
$$

Applying $F^{-1}(\theta) \succeq \frac{1}{G^2}I$ on the first inner product, and Cauchy-Schwartz on the second inner product term leads to

$$
\begin{aligned}
J^H(\theta_{t+1}^{j+1}) \geq{}& J^H(\theta_t^{j+1}) - \frac{\eta}{\mu_F}\|\nabla J^H(\theta_t^{j+1}) - u_t^{j+1}\|^2 \\
&- \frac{\mu_F}{4\eta}\|\theta_{t+1}^{j+1} - \theta_t^{j+1}\|^2 + \frac{\eta}{2G^2}\|u_t^{j+1}\|^2 \\
&- \frac{\eta}{4G^2}\|u_t^{j+1}\|^2 - \frac{G^2}{4\eta}\|\theta_{t+1}^{j+1} - \theta_{t+1,\star}^{j+1}\|^2 \\
&+ \frac{1}{2}\langle \frac{1}{\eta}F(\theta_t^{j+1})(\theta_{t+1,\star}^{j+1} - \theta_t^{j+1}), \theta_{t+1}^{j+1} - \theta_t^{j+1}\rangle - \frac{L_J}{2}\|\theta_{t+1}^{j+1} - \theta_t^{j+1}\|^2 \\
={}& J^H(\theta_t^{j+1}) - \frac{\eta}{\mu_F}\|\nabla J^H(\theta_t^{j+1}) - u_t^{j+1}\|^2 \\
&- \left(\frac{\mu_F}{4\eta} + \frac{L_J}{2}\right)\|\theta_{t+1}^{j+1} - \theta_t^{j+1}\|^2 + \frac{\eta}{4G^2}\|u_t^{j+1}\|^2 \\
&+ \frac{1}{2}\langle \frac{1}{\eta}F(\theta_t^{j+1})(\theta_{t+1,\star}^{j+1} - \theta_t^{j+1}), \theta_{t+1}^{j+1} - \theta_t^{j+1}\rangle - \frac{G^2}{4\eta}\|\theta_{t+1}^{j+1} - \theta_{t+1,\star}^{j+1}\|^2 \\
={}& J^H(\theta_t^{j+1}) \\
&- \frac{\eta}{\mu_F}\|\nabla J^H(\theta_t^{j+1}) - u_t^{j+1}\|^2 - \left(\frac{\mu_F}{4\eta} + \frac{L_J}{2}\right)\|\theta_{t+1}^{j+1} - \theta_t^{j+1}\|^2 + \frac{\eta}{4G^2}\|u_t^{j+1}\|^2 \\
&+ \frac{1}{2}\langle \frac{1}{\eta}F(\theta_t^{j+1})(\theta_{t+1}^{j+1} - \theta_t^{j+1}), \theta_{t+1}^{j+1} - \theta_t^{j+1}\rangle \\
&+ \frac{1}{2}\langle \frac{1}{\eta}F(\theta_t^{j+1})(\theta_{t+1,\star}^{j+1} - \theta_{t+1}^{j+1}), \theta_{t+1}^{j+1} - \theta_t^{j+1}\rangle \\
&- \frac{G^2}{4\eta}\|\theta_{t+1}^{j+1} - \theta_{t+1,\star}^{j+1}\|^2.
\end{aligned}
$$

Applying $\|\nabla J^H(\theta_t^{j+1})\|^2 \leq 2\|\nabla J^H(\theta_t^{j+1}) - u_t^{j+1}\|^2 + 2\|u_t^{j+1}\|^2$ and $F(\theta_t^{j+1}) \succeq \mu_F I_d$ yields

$$
\begin{aligned}
J^H(\theta_{t+1}^{j+1}) &\geq J^H(\theta_t^{j+1}) - \left(\frac{\eta}{\mu_F} + \frac{\eta}{4G^2}\right)\|\nabla J^H(\theta_t^{j+1}) - u_t^{j+1}\|^2 \\
&\quad - \left(\frac{\mu_F}{4\eta} + \frac{L_J}{2}\right)\|\theta_{t+1}^{j+1} - \theta_t^{j+1}\|^2 + \frac{\eta}{8G^2}\|\nabla J^H(\theta_t^{j+1})\|^2 \\
&\quad + \frac{\mu_F}{2\eta}\|\theta_{t+1}^{j+1} - \theta_t^{j+1}\|^2 + \frac{1}{2}\langle \frac{1}{\eta}F(\theta_t^{j+1})(\theta_{t+1,\star}^{j+1} - \theta_{t+1}^{j+1}), \theta_{t+1}^{j+1} - \theta_t^{j+1}\rangle \\
&\quad - \frac{G^2}{4\eta}\|\theta_{t+1}^{j+1} - \theta_{t+1,\star}^{j+1}\|^2 \\
&\geq J^H(\theta_t^{j+1}) - \left(\frac{\eta}{\mu_F} + \frac{\eta}{4G^2}\right)\|\nabla J^H(\theta_t^{j+1}) - u_t^{j+1}\|^2 \\
&\quad - \left(\frac{\mu_F}{4\eta} + \frac{L_J}{2}\right)\|\theta_{t+1}^{j+1} - \theta_t^{j+1}\|^2 + \frac{\eta}{8G^2}\|\nabla J^H(\theta_t^{j+1})\|^2 \qquad \text{(I.4)} \\
&\quad + \frac{\mu_F}{2\eta}\|\theta_{t+1}^{j+1} - \theta_t^{j+1}\|^2 - \frac{\mu_F^2}{2\mu_F\eta}\|\theta_{t+1,\star}^{j+1} - \theta_{t+1}^{j+1}\|^2 - \frac{\mu_F}{8\eta}\|\theta_{t+1}^{j+1} - \theta_t^{j+1}\|^2 \\
&\quad - \frac{G^2}{4\eta}\|\theta_{t+1}^{j+1} - \theta_{t+1,\star}^{j+1}\|^2 \\
&= J^H(\theta_t^{j+1}) - \left(\frac{\eta}{\mu_F} + \frac{\eta}{4G^2}\right)\|\nabla J^H(\theta_t^{j+1}) - u_t^{j+1}\|^2 \\
&\quad + \left(\frac{\mu_F}{8\eta} - \frac{L_J}{2}\right)\|\theta_{t+1}^{j+1} - \theta_t^{j+1}\|^2 + \frac{\eta}{8G^2}\|\nabla J^H(\theta_t^{j+1})\|^2 \\
&\quad - (\frac{\mu_F}{4\eta} + \frac{G^2}{4\eta})\|\theta_{t+1,\star}^{j+1} - \theta_{t+1}^{j+1}\|^2.
\end{aligned}
$$

From Lemma I.1, we further know that

$$
\begin{aligned}
\mathbb{E}[J^H(\theta_{t+1}^{j+1})] &\geq \mathbb{E}[J^H(\theta_t^{j+1})] - \left(\frac{\eta}{\mu_F} + \frac{\eta}{4G^2}\right)\left(\frac{C_\gamma}{B}\sum_{t=0}^{m-1}\mathbb{E}\|\theta_{t+1}^{j+1} - \theta_t^{j+1}\|^2 + \frac{\sigma^2}{N}\right) \\
&\quad + \left(\frac{\mu_F}{8\eta} - \frac{L_J}{2}\right)\mathbb{E}\|\theta_{t+1}^{j+1} - \theta_t^{j+1}\|^2 \\
&\quad + \frac{\eta}{8G^2}\mathbb{E}\|\nabla J^H(\theta_t^{j+1})\|^2 - (\frac{\mu_F}{4\eta} + \frac{G^2}{4\eta})\mathbb{E}\|\theta_{t+1,\star}^{j+1} - \theta_{t+1}^{j+1}\|^2.
\end{aligned}
$$

Telescoping for $s = 0, 1, ..., S-1$ and $t = 0, 1, ..., m-1$ and dividing by $Sm$ gives

$$
\begin{aligned}
&\frac{8G^2}{\eta}\left(\frac{\mu_F}{8\eta} - \frac{L_J}{2} - \left(\frac{\eta}{\mu_F} + \frac{\eta}{4G^2}\right)\frac{C_\gamma m}{B}\right)\frac{1}{Sm}\sum_{s=0}^{S-1}\sum_{t=0}^{m-1}\mathbb{E}\|\theta_{t+1}^{j+1} - \theta_t^{j+1}\|^2 \\
&+ \frac{1}{Sm}\sum_{s=0}^{S-1}\sum_{t=0}^{m-1}\mathbb{E}\|\nabla J^H(\theta_t^{j+1})\|^2 \\
&\leq \frac{8G^2}{\eta}\frac{J^\star - J^H(\theta_0)}{Sm} + \left(\frac{8G^2}{\mu_F} + 2\right)\frac{\sigma^2}{N} + \left(\frac{8G^2\mu_F}{4\eta^2} + \frac{8G^4}{4\eta^2}\right)\frac{1}{Sm}\sum_{s=0}^{S-1}\sum_{t=0}^{m-1}\mathbb{E}\|\theta_{t+1,\star}^{j+1} - \theta_{t+1}^{j+1}\|^2.
\end{aligned}
$$
(I.5)

Let us first show that the first term on the left hand side of (I.5) is non-negative. In fact, from $\eta = \frac{\mu_F}{8L_J}$ and $B = \left(\frac{\eta}{\mu_F} + \frac{\eta}{4G^2}\right)\frac{4C_\gamma m}{L_J \varepsilon^{0.25}}$ we have

$$
\frac{8G^2}{\eta}\left(\frac{\mu_F}{8\eta} - \frac{L_J}{2} - \left(\frac{\eta}{\mu_F} + \frac{\eta}{4G^2}\right)\frac{C_\gamma m}{B}\right) \geq \frac{16G^2 L_J^2}{\mu_F} > 0.
$$

Therefore, in order to have

$$\frac{1}{Sm}\sum_{s=0}^{S-1}\sum_{t=0}^{m-1}\mathbb{E}\|\nabla J^H(\theta_t^{j+1})\|^2 \le \varepsilon,$$

we can set all the three terms on the right hand side of (I.5) to be $\frac{\varepsilon}{3}$, which gives

$$Sm = \frac{24G^2(J^{H,\star} - J^H(\theta_0))}{\eta\varepsilon},$$

$$N = 3\left(\frac{8G^2}{\mu_F} + 2\right)\frac{\sigma^2}{\varepsilon}$$

$$\mathbb{E}[\|\theta_{t+1}^{j+1} - \theta_{t+1,\star}^{j+1}\|^2] \le \frac{\varepsilon}{3\left(\frac{8G^2\mu_F}{4\eta^2} + \frac{8G^4}{4\eta^2}\right)},$$

where the last requirement is satisfied according to (I.3).

For the parameters $S, m, B$, and $N$, we have

$$S = \frac{24G^2(J^{H,\star} - J^H(\theta_0))}{\eta\varepsilon^{0.5}} = \mathcal{O}\left(\frac{1}{(1-\gamma)^2\varepsilon^{0.5}}\right),$$

$$m = \frac{1}{\varepsilon^{0.5}},$$

$$B = \left(\frac{\eta}{\mu_F} + \frac{\eta}{4G^2}\right)\frac{4C_\gamma m}{L_J\varepsilon^{0.25}} = \mathcal{O}\left(\frac{W}{(1-\gamma)\varepsilon^{0.75}}\right),$$

$$N = 3\left(\frac{8G^2}{\mu_F} + 2\right)\frac{\sigma^2}{\varepsilon} = \mathcal{O}\left(\frac{\sigma^2}{\varepsilon}\right),$$

where we have applied the definition of $C_\gamma$ in Lemma I.1 in the third equality.

Therefore in total, the number of trajectories required by SRVR-NPG to reach $\varepsilon-$stationarity is

$$S\left(N + mB + (m+1)\frac{4\left((\frac{2G^2R}{\mu_F(1-\gamma)^2} + \frac{2}{1-\gamma})\sqrt{d} + \frac{2G^2R}{\mu_F(1-\gamma)^2}\right)^2}{\mu_F\frac{\varepsilon}{3\left(\frac{8G^2\mu_F}{4} + \frac{8G^4}{4}\right)}}\right)$$

$$= \mathcal{O}\left(\frac{\sigma^2}{(1-\gamma)^2\varepsilon^{1.5}} + \frac{W}{(1-\gamma)^3\varepsilon^{1.75}} + \frac{1}{(1-\gamma)^6\varepsilon^2}\right).$$

$\square$

## J   Proof of Proposition 4.5

In this section, we proceed to prove Proposition 4.5, which establishes a general global convergence result on policy gradient methods of the form $\theta^{k+1} = \theta^k + \eta w^k$.

*Proof.* First, by the $M-$smoothness of score function (see Assumption 4.2), we know that

$$\mathbb{E}_{s\sim d_\rho^{\pi^\star}}\left[\text{KL}\left(\pi^\star(\cdot\,|\,s)||\pi_{\theta^k}(\cdot\,|\,s)\right) - \text{KL}\left(\pi^\star(\cdot\,|\,s)||\pi_{\theta^{k+1}}(\cdot\,|\,s)\right)\right]$$

$$= \mathbb{E}_{s\sim d_\rho^{\pi^\star}}\mathbb{E}_{a\sim\pi^\star(\cdot\,|\,s)}\left[\log\frac{\pi_{\theta^{k+1}}(a\,|\,s)}{\pi_{\theta^k}(a\,|\,s)}\right]$$

$$\ge \mathbb{E}_{s\sim d_\rho^{\pi^\star}}\mathbb{E}_{a\sim\pi^\star(\cdot\,|\,s)}[\nabla_\theta\log\pi_{\theta^k}(a\,|\,s)\cdot(\theta^{k+1} - \theta^k)] - \frac{M}{2}\|\theta^{k+1} - \theta^k\|^2$$

$$= \eta\mathbb{E}_{s\sim d_\rho^{\pi^\star}}\mathbb{E}_{a\sim\pi^\star(\cdot\,|\,s)}[\nabla_\theta\log\pi_{\theta^k}(a\,|\,s)\cdot w^k] - \frac{M\eta^2}{2}\|w^k\|^2$$

$$= \eta\mathbb{E}_{s\sim d_\rho^{\pi^\star}}\mathbb{E}_{a\sim\pi^\star(\cdot\,|\,s)}[\nabla_\theta\log\pi_{\theta^k}(a\,|\,s)\cdot w_\star^k]$$

$$+ \eta\mathbb{E}_{s\sim d_\rho^{\pi^\star}}\mathbb{E}_{a\sim\pi^\star(\cdot\,|\,s)}[\nabla_\theta\log\pi_{\theta^k}(a\,|\,s)\cdot(w^k - w_\star^k)] - \frac{M\eta^2}{2}\|w^k\|^2$$

where we have applied KL $(p||q) = \mathbb{E}_{x \sim p}[- \log \frac{q(x)}{p(x)}]$ in the first step.

On the other hand, by the performance difference lemma [59] we know that

$$\mathbb{E}_{s \sim d_\rho^{\pi^\star}} \mathbb{E}_{a \sim \pi^\star(\cdot \mid s)}[A^{\pi_{\theta^k}}(s, a)] = (1 - \gamma)\left(J^\star - J(\theta^k)\right).$$

Therefore,

$$\mathbb{E}_{s \sim d_\rho^{\pi^\star}}\left[\mathrm{KL}\left(\pi^\star(\cdot \mid s)||\pi_{\theta^k}(\cdot \mid s)\right) - \mathrm{KL}\left(\pi^\star(\cdot \mid s)||\pi_{\theta^{k+1}}(\cdot \mid s)\right)\right]$$

$$\geq \eta \mathbb{E}_{s \sim d_\rho^{\pi^\star}} \mathbb{E}_{a \sim \pi^\star(\cdot \mid s)}[\nabla_\theta \log \pi_{\theta^k}(a \mid s) \cdot w_\star^k]$$

$$+ \eta \mathbb{E}_{s \sim d_\rho^{\pi^\star}} \mathbb{E}_{a \sim \pi^\star(\cdot \mid s)}[\nabla_\theta \log \pi_{\theta^k}(a \mid s) \cdot (w^k - w_\star^k)] - \frac{M\eta^2}{2}\|w^k\|^2$$

$$= \eta\left(J^\star - J(\theta^k)\right) + \eta \mathbb{E}_{s \sim d_\rho^{\pi^\star}} \mathbb{E}_{a \sim \pi^\star(\cdot \mid s)}[\nabla_\theta \log \pi_{\theta^k}(a \mid s) \cdot w_\star^k - \frac{1}{1 - \gamma}A^{\pi_{\theta^k}}(s, a)]$$

$$+ \eta \mathbb{E}_{s \sim d_\rho^{\pi^\star}} \mathbb{E}_{a \sim \pi^\star(\cdot \mid s)}[\nabla_\theta \log \pi_{\theta^k}(a \mid s) \cdot (w^k - w_\star^k)] - \frac{M\eta^2}{2}\|w^k\|^2$$

$$= \eta\left(J^\star - J(\theta^k)\right) + \eta\frac{1}{1 - \gamma}\mathbb{E}_{s \sim d_\rho^{\pi^\star}} \mathbb{E}_{a \sim \pi^\star(\cdot \mid s)}[\nabla_\theta \log \pi_{\theta^k}(a \mid s) \cdot (1 - \gamma)w_\star^k - A^{\pi_{\theta^k}}(s, a)]$$

$$+ \eta \mathbb{E}_{s \sim d_\rho^{\pi^\star}} \mathbb{E}_{a \sim \pi^\star(\cdot \mid s)}[\nabla_\theta \log \pi_{\theta^k}(a \mid s) \cdot (w^k - w_\star^k)] - \frac{M\eta^2}{2}\|w^k\|^2.$$

Now, let us apply Jensen's inequality and Assumption 4.2 to obtain

$$\mathbb{E}_{s \sim d_\rho^{\pi^\star}}\left[\mathrm{KL}\left(\pi^\star(\cdot \mid s)||\pi_{\theta^k}(\cdot \mid s)\right) - \mathrm{KL}\left(\pi^\star(\cdot \mid s)||\pi_{\theta^{k+1}}(\cdot \mid s)\right)\right]$$

$$\geq \eta\left(J^\star - J(\theta^k)\right)$$

$$- \eta\frac{1}{1 - \gamma}\sqrt{\mathbb{E}_{s \sim d_\rho^{\pi^\star}} \mathbb{E}_{a \sim \pi^\star(\cdot \mid s)}\left[\left(\nabla_\theta \log \pi_{\theta^k}(a \mid s) \cdot (1 - \gamma)w_\star^k - A^{\pi_{\theta^k}}(s, a)\right)^2\right]}$$

$$- \eta G\|w^k - w_\star^k\| - \frac{M\eta^2}{2}\|w^k\|^2$$

Combining this with Assumption 4.4 yields

$$\mathbb{E}_{s \sim d_\rho^{\pi^\star}}\left[\mathrm{KL}\left(\pi^\star(\cdot \mid s)||\pi_{\theta^k}(\cdot \mid s)\right) - \mathrm{KL}\left(\pi^\star(\cdot \mid s)||\pi_{\theta^{k+1}}(\cdot \mid s)\right)\right]$$

$$\geq \eta\left(J^\star - J(\theta^k)\right) - \eta\sqrt{\frac{1}{(1 - \gamma)^2}\varepsilon_{\mathrm{bias}}} - \eta G\|w^k - w_\star^k\| - \frac{M\eta^2}{2}\|w^k\|^2. \tag{J.1}$$

Finally, let us telescope the above inequality from $k = 0$ to $K - 1$, and divide by $K$, which gives

$$J(\pi^\star) - \frac{1}{K}\sum_{k=0}^{K-1} J(\theta^k) \leq \frac{\sqrt{\varepsilon_{\mathrm{bias}}}}{1 - \gamma} + \frac{1}{\eta K}\mathbb{E}_{s \sim d_\rho^{\pi^\star}}\left[\mathrm{KL}\left(\pi^\star(\cdot \mid s)||\pi_{\theta^0}(\cdot \mid s)\right)\right]$$

$$+ \frac{G}{K}\sum_{k=0}^{K-1}\|w^k - w_\star^k\| + \frac{M\eta}{2K}\sum_{k=0}^{K-1}\|w^k\|^2. \tag{J.2}$$

$\square$

On the right hand side of (J.2), the first term reflects the function approximation error due to the possibly imperfect policy parametrization. The second term vanishes as $K \to \infty$.

By looking at the third and fourth term, we know that for an update of the form $\theta^{k+1} = \theta^k + \eta w^k$, its global convergence rate depends crucially on i) the difference between its update directions $w^k$ and the exact NPG update direction $w_\star^k$, and ii) its stationary convergence rate.

In the rest of this paper, we shall see that for stochastic PG, NPG, SRVR-PG, and SRVR-NPG, both the third and fourth terms of (J.2) go to 0 as $K \to \infty$, whose speed lead to different global convergence rates for different algorithms. In order to achieve this, we will apply some intermediate results in the previous proof of stationary convergence.

## K  Proof of Theorem 4.6

Let us take $w^k$ as the update direction of PG and apply Proposition 4.5. To this end, we need to upper bound $\frac{1}{K}\sum_{k=0}^{K-1}\|w^k - w_\star^k\|$, $\frac{1}{K}\sum_{k=0}^{K-1}\|w^k\|^2$, and $\frac{1}{K}\mathbb{E}_{s\sim d_\rho^{\pi^\star}}\left[\mathrm{KL}\left(\pi^\star(\cdot\,|\,s)\|\pi_{\theta^0}(\cdot\,|\,s)\right)\right]$, where $w_\star^k = F^{-1}(\theta^k)\nabla J(\theta^k)$ is the exact NPG update direction at $\theta^k$.

- Bounding $\frac{1}{K}\sum_{k=0}^{K-1}\|w^k - w_\star^k\|$.

    We know from Jensen's inequality and $\left(\mathbb{E}[\|w_t^{j+1} - w_{t,\star}^{j+1}\|]\right)^2 \leq \mathbb{E}[\|w_t^{j+1} - w_{t,\star}^{j+1}\|^2]$ that

$$
\begin{aligned}
&\left(\frac{1}{K}\sum_{k=0}^{K-1}\mathbb{E}[\|w^k - w_\star^k\|]\right)^2 \\
&\leq \frac{1}{K}\sum_{k=0}^{K-1}\left(\mathbb{E}[\|w^k - w_\star^k\|]\right)^2 \\
&\leq \frac{1}{K}\sum_{k=0}^{K-1}\mathbb{E}[\|w^k - w_\star^k\|^2] \\
&\leq \frac{2}{K}\sum_{k=0}^{K-1}\mathbb{E}[\|w^k - \nabla J(\theta^k)\|^2] + \frac{2}{K}\sum_{k=0}^{K-1}\mathbb{E}[\|\nabla J(\theta^k) - F^{-1}(\theta^k)\nabla J(\theta^k)\|^2]
\end{aligned}
\tag{K.1}
$$

    Since

$$
w^k = \frac{1}{N}\sum_{i=1}^{N}g(\tau_i^H\,|\,\theta^k),
$$

    we have from Lemma B.1 and Assumption 4.1 that

$$
\begin{aligned}
&\frac{1}{K}\sum_{k=0}^{K-1}\mathbb{E}[\|w^k - \nabla J(\theta^k)\|^2] \\
&\leq \frac{2}{K}\sum_{k=0}^{K-1}\mathbb{E}[\|w^k - \nabla J^H(\theta^k)\|^2] + \frac{2}{K}\sum_{k=0}^{K-1}\mathbb{E}[\|\nabla J^H(\theta^k) - \nabla J(\theta^k)\|^2] \\
&\leq 2\frac{\sigma^2}{N} + 2G^2R^2\left(\frac{H+1}{1-\gamma} + \frac{\gamma}{(1-\gamma)^2}\right)^2\gamma^{2H}.
\end{aligned}
\tag{K.2}
$$

    Furthermore, Assumption 2.1 tells us that

$$
\begin{aligned}
&\mathbb{E}[\|\nabla J(\theta^k) - F^{-1}(\theta^k)\nabla J(\theta^k)\|^2] \\
&\leq \left(1 + \frac{1}{\mu_F}\right)^2\mathbb{E}[\|\nabla J(\theta^k)\|^2] \\
&\leq \left(1 + \frac{1}{\mu_F}\right)^2\left(2\mathbb{E}[\|\nabla J^H(\theta^k)\|^2] + 2G^2R^2\left(\frac{H+1}{1-\gamma} + \frac{\gamma}{(1-\gamma)^2}\right)^2\gamma^{2H}\right).
\end{aligned}
\tag{K.3}
$$

Combining (K.2) and (K.3) with (K.1) gives

$$
\frac{1}{K} \sum_{k=0}^{K-1} \mathbb{E}[\|w^k - w_\star^k\|]
$$

$$
\leq \left( 2\frac{\sigma^2}{N} + 2G^2 R^2 \left( \frac{H+1}{1-\gamma} + \frac{\gamma}{(1-\gamma)^2} \right)^2 \gamma^{2H} \right.
$$

$$
+ 2 \left( 1 + \frac{1}{\mu_F} \right)^2
$$

$$
\left. \cdot \frac{1}{K} \sum_{k=0}^{K-1} \left( 2\mathbb{E}[\|\nabla J^H(\theta^k)\|^2] + 2G^2 R^2 \left( \frac{H+1}{1-\gamma} + \frac{\gamma}{(1-\gamma)^2} \right)^2 \gamma^{2H} \right) \right)^{0.5}
$$

(K.4)

And recall from (F.2) that

$$
\frac{1}{K} \sum_{k=0}^{K-1} \mathbb{E}[\|\nabla J^H(\theta^k)\|^2] \leq \frac{\frac{J^{H,\star} - J^H(\theta_0)}{K} + (\frac{\eta}{2} + L_J \eta^2)\frac{\sigma^2}{N}}{\frac{\eta}{2} - L_J \eta^2}.
$$

Let us take $\eta = \frac{1}{4L_J}$. In addition, let $H$, $N$, and $K$ satisfy

$$
\frac{1}{3}(\frac{\varepsilon}{3G})^2 \geq \left( 2 + 4(1 + \frac{1}{\mu_F})^2 \right) G^2 R^2 \left( \frac{H+1}{1-\gamma} + \frac{\gamma}{(1-\gamma)^2} \right)^2 \gamma^{2H}
$$

$$
N \geq \frac{\left( 2 + 12(1 + \frac{1}{\mu_F})^2 \right) \sigma^2}{\frac{1}{3}\left( \frac{\varepsilon}{3G} \right)^2},
$$

(K.5)

$$
K \geq \frac{\left( 1 + \frac{1}{\mu_F} \right)^2 64 L_J (J^{H,\star} - J^H(\theta_0))}{\frac{1}{3}\left( \frac{\varepsilon}{3G} \right)^2}.
$$

Then, we have

$$
\frac{G}{K} \sum_{k=0}^{K-1} \mathbb{E}[\|w^k - w_\star^k\|] \leq \frac{\varepsilon}{3}.
$$

(K.6)

- Bounding $\frac{1}{K} \sum_{k=0}^{K-1} \|w^k\|^2$.
  We have from (K.2) and (F.2) that

$$
\frac{1}{K} \sum_{k=0}^{K-1} \mathbb{E}\|w^k\|^2
$$

$$
\leq \frac{\sigma^2}{N} + \frac{1}{K} \sum_{k=0}^{K-1} \mathbb{E}[\|\nabla J^H(\theta^k)\|^2]
$$

$$
\leq \frac{\sigma^2}{N} + \frac{\frac{J^{H,\star} - J^H(\theta_0)}{K} + (\frac{\eta}{2} + L_J \eta^2)\frac{\sigma^2}{N}}{\frac{\eta}{2} - L_J \eta^2}.
$$

Taking $\eta = \frac{1}{4L_j}$ and

$$
N \geq \frac{12M\eta\sigma^2}{\varepsilon},
$$

(K.7)

$$
K \geq \frac{48L_J M\eta(J^{H,\star} - J^H(\theta_0))}{\varepsilon},
$$

we arrive at

$$
\frac{M\eta}{2K} \sum_{k=0}^{K-1} \mathbb{E}[\|w^k\|^2] \leq \frac{\varepsilon}{3}.
$$

(K.8)

- Bounding $\frac{1}{K}\mathbb{E}_{s\sim d_\rho^{\pi^\star}}\left[\text{KL}\left(\pi^\star(\cdot\,|\,s)||\pi_{\theta^0}(\cdot\,|\,s)\right)\right]$.

  By taking

$$K \geq \frac{3\mathbb{E}_{s\sim d_\rho^{\pi^\star}}\left[\text{KL}\left(\pi^\star(\cdot\,|\,s)||\pi_{\theta^0}(\cdot\,|\,s)\right)\right]}{\eta\varepsilon} \tag{K.9}$$

  we have

$$\frac{1}{\eta K}\mathbb{E}_{s\sim d_\rho^{\pi^\star}}\left[\text{KL}\left(\pi^\star(\cdot\,|\,s)||\pi_{\theta^0}(\cdot\,|\,s)\right)\right] \leq \frac{\varepsilon}{3}, \tag{K.10}$$

In summary, we require $N$ and $K$ to satisfy (K.5), (K.7), and (K.9), which leads to

$$N = \mathcal{O}\left(\frac{\sigma^2}{\varepsilon^2}\right), \qquad K = \mathcal{O}\left(\frac{1}{(1-\gamma)^2\varepsilon^2}\right), \qquad H = \mathcal{O}\left(\log((1-\gamma)^{-1}\varepsilon^{-1})\right).$$

By combining (K.6), (K.8), (K.10) and (J.2), we can conclude that

$$J(\pi^\star) - \frac{1}{K}\sum_{k=0}^{K-1} J(\theta^k) \leq \frac{\sqrt{\varepsilon_{\text{bias}}}}{1-\gamma} + \varepsilon.$$

In total, stochastic PG requires to sample $KN = \mathcal{O}\left(\frac{\sigma^2}{(1-\gamma)^2\varepsilon^4}\right)$ trajectories.

## L  Proof of Theorem 4.9

Let us take $w^k$ as the update direction of NPG and apply Proposition 4.5. To this end, we need to upper bound $\frac{1}{K}\sum_{k=0}^{K-1}\|w^k - w_\star^k\|$, $\frac{1}{K}\sum_{k=0}^{K-1}\|w^k\|^2$, and $\frac{1}{K}\mathbb{E}_{s\sim d_\rho^{\pi^\star}}\left[\text{KL}\left(\pi^\star(\cdot\,|\,s)||\pi_{\theta^0}(\cdot\,|\,s)\right)\right]$, where $w_\star^k = F^{-1}(\theta^k)\nabla J(\theta^k)$ is the exact NPG update direction at $\theta^k$.

Let us take $\eta = \frac{\mu_F^2}{4G^2 L_J}$ and apply SGD as in Procedure 1 to obtain a $w^k$ that satisfies

$$\mathbb{E}[\|w^k - w_\star^k\|^2] \leq \min\left\{\frac{\varepsilon}{12M\eta}, \left(\frac{1}{G}\frac{\varepsilon}{3}\right)^2, \frac{\varepsilon}{12M\eta^3}\frac{\mu_F^2}{2}\frac{\frac{\eta}{2G^2} - \frac{L_J\eta^2}{\mu_F^2}}{\frac{G^2}{2\eta} + L_J}\right\}. \tag{L.1}$$

From Proposition G.1, we know that this requires sampling $\mathcal{O}\left(\frac{1}{(1-\gamma)^4\varepsilon^2}\right)$ trajectories at each iteration.

- Bounding $\frac{1}{K}\sum_{k=0}^{K-1}\|w^k - w_\star^k\|$.

  Recall that the update direction $w^k \approx w_\star^k = F^{-1}(\theta^k)\nabla J(\theta^k)$ is obtained by solving the subproblem

$$w^k \approx \underset{w\in\mathbb{R}^d}{\arg\min}\, L_{\nu^{\pi_\theta}}(w;\theta^k)$$

$$= \underset{w\in\mathbb{R}^d}{\arg\min}\, \mathbb{E}_{(s,a)\sim\nu^{\pi_{\theta^k}}}\left[A^{\pi_{\theta^k}}(s,a) - (1-\gamma)w^\top\nabla_\theta\log\pi_{\theta^k}(a\,|\,s)\right]^2.$$

  By (L.1) and Jensen's inequality, we can write

$$\left(\frac{1}{K}\sum_{k=0}^{K-1}\mathbb{E}[\|w^k - w_\star^k\|]\right)^2 \leq \frac{1}{K}\sum_{k=0}^{K-1}\mathbb{E}[\|w^k - w_\star^k\|^2] \leq \left(\frac{1}{G}\frac{\varepsilon}{3}\right)^2. \tag{L.2}$$

  On the other hand, by replacing (G.1) with (L.1), the stationary convergence of NPG stated in (G.3) becomes

$$\frac{1}{K}\sum_{k=0}^{K-1}\mathbb{E}[\|\nabla J(\theta^k)\|^2] \leq \frac{\frac{J^\star - J(\theta_0)}{K} + \frac{\varepsilon}{12M\eta}\frac{\mu_F^2}{2}\left(\frac{\eta}{2G^2} - \frac{L_J\eta^2}{\mu_F^2}\right)}{\frac{\eta}{2G^2} - \frac{L_J\eta^2}{\mu_F^2}}$$

- Bounding $\frac{1}{K} \sum_{k=0}^{K-1} \|w^k\|^2$.

  Taking $\eta = \frac{\mu_F^2}{4G^2 L_J}$ and

  $$K \geq \frac{24(J^\star - J(\theta_0))M\eta}{\mu_F^2 \left( \frac{\eta}{2G^2} - \frac{L_J \eta^2}{\mu_F^2} \right) \varepsilon} \tag{L.3}$$

  gives us

  $$\frac{1}{K} \sum_{k=0}^{K-1} \mathbb{E}[\|\nabla J(\theta^k)\|^2] \leq \frac{\mu_F^2 \varepsilon}{12M\eta}.$$

  (L.1) and the above inequality yields

  $$\frac{1}{K} \sum_{k=0}^{K-1} \mathbb{E}[\|w^k\|^2] \leq \frac{2}{K} \sum_{k=0}^{K-1} \mathbb{E}[\|w^k - w_\star^k\|^2] + \frac{1}{\mu_F^2} \frac{2}{K} \sum_{k=0}^{K-1} \mathbb{E}[\|\nabla J(\theta^k)\|^2] \tag{L.4}$$
  $$\leq \frac{2\varepsilon}{12M\eta} + \frac{2}{\mu_F^2} \cdot \frac{\mu_F^2 \varepsilon}{12M\eta} = \frac{\varepsilon}{3M\eta}.$$

- Bounding $\frac{1}{K} \mathbb{E}_{s \sim d_\rho^{\pi^\star}} \left[ \text{KL} \left( \pi^\star(\cdot \,|\, s) || \pi_{\theta^0}(\cdot \,|\, s) \right) \right]$.

  Let us also set

  $$K \geq \frac{3\mathbb{E}_{s \sim d_\rho^{\pi^\star}} \left[ \text{KL} \left( \pi^\star(\cdot \,|\, s) || \pi_{\theta^0}(\cdot \,|\, s) \right) \right]}{\eta \varepsilon}, \tag{L.5}$$

  so that

  $$\frac{1}{\eta K} \mathbb{E}_{s \sim d_\rho^{\pi^\star}} \left[ \text{KL} \left( \pi^\star(\cdot \,|\, s) || \pi_{\theta^0}(\cdot \,|\, s) \right) \right] \leq \frac{\varepsilon}{3}. \tag{L.6}$$

In summary, we require $K$ to satisfy (L.3) and (L.5), which leads to

$$K = \mathcal{O} \left( \frac{1}{(1-\gamma)^2 \varepsilon} \right)$$

By combining (L.2), (L.4), (L.6) and (J.2), we can conclude that

$$J(\pi^\star) - \frac{1}{K} \sum_{k=0}^{K-1} J(\theta^k) \leq \frac{\sqrt{\varepsilon_{\text{bias}}}}{1-\gamma} + \varepsilon.$$

Since at each iteration, SGD needs to sample $\mathcal{O} \left( \frac{1}{(1-\gamma)^4 \varepsilon^2} \right)$ trajectories so that (L.1) is satisfied, NPG requires to sample $\mathcal{O} \left( \frac{1}{(1-\gamma)^6 \varepsilon^3} \right)$ trajectories in total.

## M  Proof of Theorem 4.11

Let us take $w_t^{j+1}$ as the update direction of SRVR-PG and apply Proposition 4.5. To this end, we need to upper bound $\frac{1}{Sm} \sum_{s=0}^{S-1} \sum_{t=0}^{m-1} \|w_t^{j+1} - w_{t,\star}^{j+1}\|$, $\frac{1}{Sm} \sum_{s=0}^{S-1} \sum_{t=0}^{m-1} \|w_t^{j+1}\|^2$, and $\frac{1}{Sm} \mathbb{E}_{s \sim d_\rho^{\pi^\star}} \left[ \text{KL} \left( \pi^\star(\cdot \,|\, s) || \pi_{\theta^0}(\cdot \,|\, s) \right) \right]$, where $w_{t,\star}^{j+1} = F^{-1}(\theta_t^{j+1}) \nabla J(\theta_t^{j+1})$ is the exact NPG update direction at $\theta_t^{j+1}$.

- Bounding $\frac{1}{Sm} \sum_{s=0}^{S-1} \sum_{t=0}^{m-1} \|w_t^{j+1} - w_{t,\star}^{j+1}\|$.

Since $w_t^{j+1} = u_t^{j+1}$ and $w_{t,\star}^{j+1} = F^{-1}(\theta_t^{j+1})\nabla J(\theta_t^{j+1})$, we have from Lemmas I.1 and B.1 that

$$\mathbb{E}[\|w_t^{j+1} - w_{t,\star}^{j+1}\|^2]$$

$$\leq 2\mathbb{E}[\|u_t^{j+1} - \nabla J(\theta_t^{j+1})\|^2] + 2\mathbb{E}[\|\nabla J(\theta_t^{j+1}) - F^{-1}(\theta_t^{j+1})\nabla J(\theta_t^{j+1})\|^2]$$

$$\leq 2\mathbb{E}[\|u_t^{j+1} - \nabla J^H(\theta_t^{j+1})\|^2] + 2(1 + \frac{1}{\mu_F})^2\mathbb{E}[\|\nabla J(\theta_t^{j+1})\|^2]$$

$$+ 2G^2R^2\left(\frac{H+1}{1-\gamma} + \frac{\gamma}{(1-\gamma)^2}\right)^2 \gamma^{2H}$$

$$\leq 2\left(\frac{C_\gamma}{B}\sum_{l=1}^{t}\mathbb{E}[\|\theta_l^{j+1} - \theta_{l-1}^{j+1}\|^2] + \frac{\sigma^2}{N}\right) + 2(1 + \frac{1}{\mu_F})^2\mathbb{E}[\|\nabla J(\theta_t^{j+1})\|^2]$$

$$+ 2G^2R^2\left(\frac{H+1}{1-\gamma} + \frac{\gamma}{(1-\gamma)^2}\right)^2 \gamma^{2H}$$

$$\leq 2\left(\frac{C_\gamma}{B}\sum_{t=0}^{m-1}\mathbb{E}[\|\theta_{t+1}^{j+1} - \theta_t^{j+1}\|^2] + \frac{\sigma^2}{N}\right) + 2(1 + \frac{1}{\mu_F})^2\mathbb{E}[\|\nabla J(\theta_t^{j+1})\|^2]$$

$$+ 2G^2R^2\left(\frac{H+1}{1-\gamma} + \frac{\gamma}{(1-\gamma)^2}\right)^2 \gamma^{2H},$$

where we have applied Lemma B.1 and Assumption 2.1 in the second inequality, and Lemma I.1 in the third one.

Telescoping this over $s = 0, 1, .., S-1$, $t = 0, 1, m-1$ and dividing by $Sm$ gives

$$\frac{1}{Sm}\sum_{s=0}^{S-1}\sum_{t=0}^{m-1}\mathbb{E}[\|w_t^{j+1} - w_{t,\star}^{j+1}\|^2]$$

$$\leq 2(1 + \frac{1}{\mu_F})^2\frac{1}{Sm}\sum_{s=0}^{S-1}\sum_{t=0}^{m-1}\mathbb{E}[\|\nabla J(\theta_t^{j+1})\|^2]$$

$$+ 2\left(\frac{C_\gamma m}{B}\frac{1}{Sm}\sum_{s=0}^{S-1}\sum_{t=0}^{m-1}\mathbb{E}[\|\theta_{t+1}^{j+1} - \theta_t^{j+1}\|^2] + \frac{\sigma^2}{N}\right) \tag{M.1}$$

$$+ 2G^2R^2\left(\frac{H+1}{1-\gamma} + \frac{\gamma}{(1-\gamma)^2}\right)^2 \gamma^{2H}.$$

On the other hand, from Equation (B.14) of [21] we know that

$$\left(\frac{2}{\eta^2} - \frac{4L_J}{\eta} - \frac{12mC_\gamma}{B}\right)\frac{1}{Sm}\sum_{s=0}^{S-1}\sum_{t=0}^{m-1}\mathbb{E}[\|\theta_{t+1}^{j+1} - \theta_t^{j+1}\|^2]$$

$$+ \frac{1}{Sm}\sum_{s=0}^{S-1}\sum_{t=0}^{m-1}\mathbb{E}[\|\nabla J^H(\theta_t^{j+1})\|^2] \tag{M.2}$$

$$\leq \frac{8(J^{H,\star} - J^H(\theta_0))}{\eta Sm} + \frac{6\sigma^2}{N}.$$

By the definition of $C_\gamma$ in Lemma I.1, we have

$$B = \frac{3\eta C_\gamma m}{L_J} = \frac{72\eta RG(2G^2 + M)(W+1)\gamma}{L_J(1-\gamma)^5}m \tag{M.3}$$

Therefore, (M.2) becomes

$$\left(\frac{2}{\eta^2} - \frac{8L_J}{\eta}\right)\frac{1}{Sm}\sum_{s=0}^{S-1}\sum_{t=0}^{m-1}\mathbb{E}[\|\theta_{t+1}^{j+1} - \theta_t^{j+1}\|^2] + \frac{1}{Sm}\sum_{s=0}^{S-1}\sum_{t=0}^{m-1}\mathbb{E}[\|\nabla J^H(\theta_t^{j+1})\|^2]$$

$$\leq \frac{8(J^{H,\star} - J^H(\theta_0))}{\eta Sm} + \frac{6\sigma^2}{N},$$

Since $\eta = \frac{1}{8L_J}$, we further have

$$\frac{1}{Sm}\sum_{s=0}^{S-1}\sum_{t=0}^{m-1}\mathbb{E}[\|\theta_{t+1}^{j+1} - \theta_t^{j+1}\|^2] \leq \frac{J^{H,\star} - J^H(\theta_0)}{L_J Sm} + \frac{6\sigma^2}{64L_J^2 N},$$

$$\frac{1}{Sm}\sum_{s=0}^{S-1}\sum_{t=0}^{m-1}\mathbb{E}[\|\nabla J^H(\theta_t^{j+1})\|^2] \leq \frac{64L_J(J^{H,\star} - J^H(\theta_0))}{Sm} + \frac{6\sigma^2}{N}. \tag{M.4}$$

Putting these inequalities back into (M.1) yields

$$\frac{1}{Sm}\sum_{s=0}^{S-1}\sum_{t=0}^{m-1}\mathbb{E}[\|w_t^{j+1} - w_{t,\star}^{j+1}\|^2] \leq 2(1 + \frac{1}{\mu_F})^2\left(\frac{64L_J(J^{H,\star} - J^H(\theta_0))}{Sm} + \frac{6\sigma^2}{N}\right)$$

$$+ 2\left(\frac{8L_J^2}{3}\left(\frac{J^{H,\star} - J^H(\theta_0)}{L_J Sm} + \frac{6\sigma^2}{64L_J^2 N}\right) + \frac{\sigma^2}{N}\right)$$

$$+ 2G^2R^2\left(\frac{H+1}{1-\gamma} + \frac{\gamma}{(1-\gamma)^2}\right)^2\gamma^{2H}.$$

Let us set

$$\frac{1}{3}\left(\frac{\varepsilon}{3G}\right)^2 \geq 2G^2R^2\left(\frac{H+1}{1-\gamma} + \frac{\gamma}{(1-\gamma)^2}\right)^2\gamma^{2H},$$

$$N \geq \frac{\left(12(1 + \frac{1}{\mu_F})^2 + 2.5\right)\sigma^2}{\frac{1}{3}\left(\frac{\varepsilon}{3G}\right)^2}, \tag{M.5}$$

$$Sm \geq \frac{\left(128(1 + \frac{1}{\mu_F})^2 + \frac{16}{3}\right)L_J(J^{H,\star} - J^H(\theta_0))}{\frac{1}{3}\left(\frac{\varepsilon}{3G}\right)^2},$$

so that

$$\frac{1}{Sm}\sum_{s=0}^{S-1}\sum_{t=0}^{m-1}\mathbb{E}[\|w_t^{j+1} - w_{t,\star}^{j+1}\|^2] \leq \left(\frac{1}{G}\frac{\varepsilon}{3}\right)^2$$

By Jensen's inequality, we further have

$$\left(\frac{1}{Sm}\sum_{s=0}^{S-1}\sum_{t=0}^{m-1}\mathbb{E}[\|w_t^{j+1} - w_{t,\star}^{j+1}\|]\right)^2 \leq \frac{1}{Sm}\sum_{s=0}^{S-1}\sum_{t=0}^{m-1}\left(\mathbb{E}[\|w_t^{j+1} - w_{t,\star}^{j+1}\|]\right)^2$$

$$\leq \frac{1}{Sm}\sum_{s=0}^{S-1}\sum_{t=0}^{m-1}\mathbb{E}[\|w_t^{j+1} - w_{t,\star}^{j+1}\|^2] \tag{M.6}$$

$$\leq \left(\frac{1}{G}\frac{\varepsilon}{3}\right)^2.$$

where we have also applied $\left(\mathbb{E}[\|w_t^{j+1} - w_{t,\star}^{j+1}\|]\right)^2 \leq \mathbb{E}[\|w_t^{j+1} - w_{t,\star}^{j+1}\|^2]$.

- Bounding $\frac{1}{Sm}\sum_{s=0}^{S-1}\sum_{t=0}^{m-1}\|w_t^{j+1}\|^2$

Since $\mathbb{E}[w_t^{j+1}] = \mathbb{E}[u_t^{j+1}] = \nabla J^H(\theta_t^{j+1})$, by Lemma I.1 we have

$$\frac{1}{Sm} \sum_{s=0}^{S-1} \sum_{t=0}^{m-1} \mathbb{E}[\|w_t^{j+1}\|^2]$$

$$= \frac{1}{Sm} \sum_{s=0}^{S-1} \sum_{t=0}^{m-1} \mathbb{E}[\|u_t^{j+1} - \nabla J^H(\theta_t^{j+1})\|^2] + \frac{1}{Sm} \sum_{s=0}^{S-1} \sum_{t=0}^{m-1} \mathbb{E}[\|\nabla J^H(\theta_t^{j+1})\|^2]$$

$$\leq \frac{1}{Sm} \sum_{s=0}^{S-1} \sum_{t=0}^{m-1} \left( \frac{C_\gamma}{B} \sum_{l=1}^{t} \mathbb{E}[\|\theta_l^{j+1} - \theta_{l-1}^{j+1}\|^2] + \frac{\sigma^2}{N} \right)$$

$$+ \frac{1}{Sm} \sum_{s=0}^{S-1} \sum_{t=0}^{m-1} \mathbb{E}[\|\nabla J^H(\theta_t^{j+1})\|^2]$$

$$\leq \frac{C_\gamma m}{B} \cdot \frac{1}{Sm} \sum_{s=0}^{S-1} \sum_{t=0}^{m-1} \mathbb{E}[\|\theta_{t+1}^{j+1} - \theta_t^{j+1}\|^2] + \frac{\sigma^2}{N} + \frac{1}{Sm} \sum_{s=0}^{S-1} \sum_{t=0}^{m-1} \mathbb{E}[\|\nabla J^H(\theta_t^{j+1})\|^2].$$

By setting $\eta = \frac{1}{8L_J}$ and applying (M.3) and (M.4), we further have

$$\frac{1}{Sm} \sum_{s=0}^{S-1} \sum_{t=0}^{m-1} \mathbb{E}[\|w_t^{j+1}\|^2]$$

$$\leq \frac{8L_J^2}{3} \cdot \left( \frac{J^{H,\star} - J^H(\theta_0)}{L_J Sm} + \frac{6\sigma^2}{64L_J^2 N} \right) + \frac{\sigma^2}{N} + \frac{64L_J(J^{H,\star} - J^H(\theta_0))}{Sm} + \frac{6\sigma^2}{N}$$

Therefore, we can set

$$N \geq \frac{174M\sigma^2}{32L_J\varepsilon},$$
$$Sm \geq \frac{50M(J^{H,\star} - J^H(\theta_0))}{\varepsilon},$$

(M.7)

so that

$$\frac{1}{Sm} \sum_{s=0}^{S-1} \sum_{t=0}^{m-1} \mathbb{E}\|w_t^{j+1}\|^2 \leq \frac{\varepsilon}{3M\eta}.$$

(M.8)

- Bounding $\frac{1}{Sm} \mathbb{E}_{s \sim d_\rho^{\pi^\star}} \left[ \mathrm{KL} \left( \pi^\star(\cdot \,|\, s) || \pi_{\theta^0}(\cdot \,|\, s) \right) \right]$.

  Let us set

  $$Sm \geq \frac{3\mathbb{E}_{s \sim d_\rho^{\pi^\star}} \left[ \mathrm{KL} \left( \pi^\star(\cdot \,|\, s) || \pi_{\theta^0}(\cdot \,|\, s) \right) \right]}{\eta\varepsilon}$$

  (M.9)

  so that

  $$\frac{1}{\eta Sm} \mathbb{E}_{s \sim d_\rho^{\pi^\star}} \left[ \mathrm{KL} \left( \pi^\star(\cdot \,|\, s) || \pi_{\theta^0}(\cdot \,|\, s) \right) \right] \leq \frac{\varepsilon}{3}.$$

  (M.10)

By combining (M.6), (M.8), (M.10) and (J.2), we can conclude that

$$J(\pi^\star) - \frac{1}{K} \sum_{k=0}^{K-1} J(\theta^k) \leq \frac{\sqrt{\varepsilon_{\text{bias}}}}{1 - \gamma} + \varepsilon.$$

To achieve this, we require $Sm$ and $N$ to satisfy (M.5), (M.7), and (M.9), which leads to

$$Sm = \mathcal{O}\left( \frac{1}{(1-\gamma)^2\varepsilon^2} \right), \qquad N = \mathcal{O}\left( \frac{\sigma^2}{\varepsilon^2} \right), \qquad H = \mathcal{O}\left( \log(\frac{1}{(1-\gamma)\varepsilon}) \right).$$

By (M.3), we know that $B = \mathcal{O}(W(1-\gamma)^{-1}m)$.

Therefore, by taking $S = \mathcal{O}\left(\frac{1}{(1-\gamma)^{2.5}\varepsilon}\right)$ and $m = \mathcal{O}\left(\frac{1}{(1-\gamma)^{-0.5}\varepsilon}\right)$, the sample complexity of SRVR-PG is

$$S(N + mB) = \mathcal{O}\left(\frac{\sigma^2}{(1-\gamma)^{2.5}\varepsilon^3} + \frac{W}{(1-\gamma)^{2.5}\varepsilon^3}\right)$$

$$= \mathcal{O}\left(\frac{W + \sigma^2}{(1-\gamma)^{2.5}\varepsilon^3}\right).$$

# N   Proof of Theorem 4.13

Let us take $w_t^{j+1}$ as the update direction of SRVR-NPG and apply Proposition 4.5. To this end, we need to upper bound $\frac{1}{Sm}\sum_{s=0}^{S-1}\sum_{t=0}^{m-1}\|w_t^{j+1} - w_{t,\star}^{j+1}\|$, $\frac{1}{Sm}\sum_{s=0}^{S-1}\sum_{t=0}^{m-1}\|w_t^{j+1}\|^2$, and $\frac{1}{Sm}\mathbb{E}_{s\sim d_\rho^{\pi^\star}}\left[\text{KL}\left(\pi^\star(\cdot\,|\,s)||\pi_{\theta^0}(\cdot\,|\,s)\right)\right]$, where $w_{t,\star}^{j+1} = F^{-1}(\theta_t^{j+1})\nabla J(\theta_t^{j+1})$ is the exact NPG update direction at $\theta_t^{j+1}$.

Let us take $\eta = \frac{\mu_F}{16L_J}$ and apply SGD as in Procedure 2 to obtain a $w_t^{j+1}$ that satisfies

$$\mathbb{E}[\|w_t^{j+1} - F^{-1}(\theta_t^{j+1})u_t^{j+1}\|^2] \le \min\left\{\frac{\frac{1}{4}\left(\frac{1}{G}\frac{\varepsilon}{3}\right)^2}{2 + \frac{G^2\mu_F + G^4}{\mu_F^2}\cdot\frac{\mu_F}{4G^2+\mu_F}}, \frac{64\eta^2 L_J^2}{\mu_F(\mu_F + G^2)}\cdot\frac{\varepsilon}{9M\eta}\right\}. \tag{N.1}$$

In order to apply Proposition I.2, let assume the following so that its assumptions are satisfied:

$$\frac{\sigma^2}{N} \le \left(\frac{GR}{(1-\gamma)^2}\right)^2,$$

$$\min\left\{\frac{\frac{1}{4}\left(\frac{1-\gamma}{G}\frac{\varepsilon}{3}\right)^2}{2 + \frac{G^2\mu_F + G^4}{\mu_F^2}\cdot\frac{\mu_F}{4G^2+\mu_F}}, \frac{64\eta^2 L_J^2}{\mu_F(\mu_F + G^2)}\cdot\frac{(1-\gamma)\varepsilon}{9M\eta}\right\} \le \frac{2}{\mu_F^2}\left(\frac{GR}{(1-\gamma)^2}\right)^2 \tag{N.2}$$

$$\frac{C_\gamma m}{B}2\eta^2 \le \frac{1}{3}\mu_F^2.$$

At the end of this proof, we will see that these assumptions are indeed satisfied for small $\varepsilon$.

From Proposition I.2, we know that this requires sampling $\mathcal{O}\left(\frac{1}{(1-\gamma)^4\varepsilon^2}\right)$ trajectories at each iteration.

- Bounding $\frac{1}{Sm}\sum_{s=0}^{S-1}\sum_{t=0}^{m-1}\|w_t^{j+1} - w_{t,\star}^{j+1}\|$.
  First of all, we have

  $\mathbb{E}[\|w_t^{j+1} - w_{t,\star}^{j+1}\|^2]$

  $\le 2\mathbb{E}[\|w_t^{j+1} - F^{-1}(\theta_t^{j+1})u_t^{j+1}\|^2] + 2\mathbb{E}[\|F^{-1}(\theta_t^{j+1})u_t^{j+1} - F^{-1}(\theta_t^{j+1})\nabla J(\theta_t^{j+1})\|^2]$

  $\le 2\mathbb{E}[\|w_t^{j+1} - F^{-1}(\theta_t^{j+1})u_t^{j+1}\|^2] + 2\frac{1}{\mu_F^2}\mathbb{E}[\|u_t^{j+1} - \nabla J(\theta_t^{j+1})\|^2]$

  $\le 2\mathbb{E}[\|w_t^{j+1} - F^{-1}(\theta_t^{j+1})u_t^{j+1}\|^2] + 4\frac{1}{\mu_F^2}\left(\frac{C_\gamma}{B}\sum_{l=1}^{t}\mathbb{E}[\|\theta_l^{j+1} - \theta_{l-1}^{j+1}\|^2] + \frac{\sigma^2}{N}\right)$

  $+ 4G^2 R^2\left(\frac{H+1}{1-\gamma} + \frac{\gamma}{(1-\gamma)^2}\right)^2\gamma^{2H}$

  $\le 2\mathbb{E}[\|w_t^{j+1} - F^{-1}(\theta_t^{j+1})u_t^{j+1}\|^2] + 4\frac{1}{\mu_F^2}\left(\frac{C_\gamma}{B}\sum_{t=0}^{m-1}\mathbb{E}[\|\theta_{t+1}^{j+1} - \theta_t^{j+1}\|^2] + \frac{\sigma^2}{N}\right)$

  $+ 4G^2 R^2\left(\frac{H+1}{1-\gamma} + \frac{\gamma}{(1-\gamma)^2}\right)^2\gamma^{2H},$

  where we have applied Assumption 2.1 in the second inequality, and Lemmas I.1 and B.1 in the third one.

Telescoping this over $s = 0, 1, .., S - 1, t = 0, 1, m - 1$ and dividing by $Sm$ gives

$$\frac{1}{Sm} \sum_{s=0}^{S-1} \sum_{t=0}^{m-1} \mathbb{E}[\|w_t^{j+1} - w_{t,\star}^{j+1}\|^2]$$

$$\leq 2 \frac{\frac{1}{4} \left(\frac{1}{G} \frac{\varepsilon}{3}\right)^2}{2 + \frac{G^2 \mu_F + G^4}{\mu_F^2} \cdot \frac{\mu_F}{4G^2 + \mu_F}} + 4 \frac{1}{\mu_F^2} \left(\frac{C_\gamma m}{B} \frac{1}{Sm} \sum_{s=0}^{S-1} \sum_{t=0}^{m-1} \mathbb{E}[\|\theta_{t+1}^{j+1} - \theta_t^{j+1}\|^2] + \frac{\sigma^2}{N}\right)$$

$$+ 4G^2 R^2 \left(\frac{H+1}{1-\gamma} + \frac{\gamma}{(1-\gamma)^2}\right)^2 \gamma^{2H}.$$

$$\tag{N.3}$$

On the other hand, from (I.5) we know that

$$\frac{8G^2}{\eta} \left(\frac{\mu_F}{8\eta} - \frac{L_J}{2} - \left(\frac{\eta}{\mu_F} + \frac{\eta}{4G^2}\right) \frac{C_\gamma m}{B}\right) \frac{1}{Sm} \sum_{s=0}^{S-1} \sum_{t=0}^{m-1} \mathbb{E}\|\theta_{t+1}^{j+1} - \theta_t^{j+1}\|^2$$

$$+ \frac{1}{Sm} \sum_{s=0}^{S-1} \sum_{t=0}^{m-1} \mathbb{E}\|\nabla J^H(\theta_t^{j+1})\|^2$$

$$\leq \frac{8G^2}{\eta} \frac{J^{H,\star} - J^H(\theta_0)}{Sm}$$

$$+ \left(\frac{8G^2}{\mu_F} + 2\right) \frac{\sigma^2}{N} + \left(\frac{8G^2 \mu_F}{4} + \frac{8G^4}{4}\right) \frac{1}{Sm} \sum_{s=0}^{S-1} \sum_{t=0}^{m-1} \mathbb{E}\|F^{-1}(\theta_t^{j+1}) u_t^{j+1} - w_t^{j+1}\|^2.$$

$$\tag{N.4}$$

Let us set

$$B \geq \left(\frac{\eta}{\mu_F} + \frac{\eta}{4G^2}\right) \frac{2C_\gamma m}{L_J} = \left(\frac{\eta}{\mu_F} + \frac{\eta}{4G^2}\right) \cdot \frac{48RG^2(2G^2 + M)(W+1)\gamma}{L_J(1-\gamma)^5} m. \tag{N.5}$$

Since $\eta = \frac{\mu_F}{16 L_J}$, (N.4) becomes

$$\frac{8G^2 L_J}{\eta} \frac{1}{Sm} \sum_{s=0}^{S-1} \sum_{t=0}^{m-1} \mathbb{E}\|\theta_{t+1}^{j+1} - \theta_t^{j+1}\|^2 + \frac{1}{Sm} \sum_{s=0}^{S-1} \sum_{t=0}^{m-1} \mathbb{E}\|\nabla J^H(\theta_t^{j+1})\|^2$$

$$\leq \frac{8G^2}{\eta} \frac{J^{H,\star} - J^H(\theta_0)}{Sm} + \left(\frac{8G^2}{\mu_F} + 2\right) \frac{\sigma^2}{N}$$

$$+ \left(\frac{8G^2 \mu_F}{4} + \frac{8G^4}{4}\right) \frac{1}{Sm} \sum_{s=0}^{S-1} \sum_{t=0}^{m-1} \mathbb{E}\|F^{-1}(\theta_t^{j+1}) u_t^{j+1} - w_t^{j+1}\|^2,$$

from which we have

$$\frac{1}{Sm} \sum_{s=0}^{S-1} \sum_{t=0}^{m-1} \mathbb{E}[\|\theta_{t+1}^{j+1} - \theta_t^{j+1}\|^2]$$

$$\leq \frac{J^{H,\star} - J^H(\theta_0)}{L_J Sm} + \left(\frac{8G^2}{\mu_F} + 2\right) \frac{\mu_F}{128 G^2 L_J^2} \frac{\sigma^2}{N}$$

$$+ \left(\frac{8G^2 \mu_F}{4} + \frac{8G^4}{4}\right) \frac{\mu_F}{128 G^2 L_J^2} \frac{1}{Sm} \sum_{s=0}^{S-1} \sum_{t=0}^{m-1} \mathbb{E}\|F^{-1}(\theta_t^{j+1}) u_t^{j+1} - w_t^{j+1}\|^2.$$

$$\tag{N.6}$$

Putting these inequalities back into (N.3) and applying (N.1) yields

$$
\frac{1}{Sm}\sum_{s=0}^{S-1}\sum_{t=0}^{m-1}\mathbb{E}[\|w_t^{j+1}-w_{t,\star}^{j+1}\|^2]
$$

$$
\leq 2\frac{\frac{1}{4}\left(\frac{1}{G}\frac{\varepsilon}{3}\right)^2}{2+\frac{G^2\mu_F+G^4}{\mu_F^2}\cdot\frac{\mu_F}{4G^2+\mu_F}}
$$

$$
+4\frac{1}{\mu_F^2}\left(\frac{C_\gamma m}{B}\frac{1}{Sm}\sum_{s=0}^{S-1}\sum_{t=0}^{m-1}\mathbb{E}[\|\theta_{t+1}^{j+1}-\theta_t^{j+1}\|^2]+\frac{\sigma^2}{N}\right)
$$

$$
+4G^2R^2\left(\frac{H+1}{1-\gamma}+\frac{\gamma}{(1-\gamma)^2}\right)^2\gamma^{2H}
$$

$$
\leq 2\frac{\frac{1}{4}\left(\frac{1}{G}\frac{\varepsilon}{3}\right)^2}{2+\frac{G^2\mu_F+G^4}{\mu_F^2}\cdot\frac{\mu_F}{4G^2+\mu_F}}
$$

$$
+2\frac{1}{\mu_F^2}\frac{C_\gamma m}{B}\left(\frac{J^\star-J(\theta_0)}{L_JSm}+\left(\frac{8G^2}{\mu_F}+2\right)\frac{\mu_F}{128G^2L_J^2}\frac{\sigma^2}{N}\right.
$$

$$
\left.+(G^2\mu_F+G^4)\frac{\mu_F}{64G^2L_J^2}\frac{\frac{1}{3}\left(\frac{1-\gamma}{G}\frac{\varepsilon}{3}\right)^2}{1+\frac{G^2\mu_F+G^4}{\mu_F^2}\cdot\frac{\mu_F}{4G^2+\mu_F}}\right)
$$

$$
+2\frac{1}{\mu_F^2}\frac{\sigma^2}{N}+4G^2R^2\left(\frac{H+1}{1-\gamma}+\frac{\gamma}{(1-\gamma)^2}\right)^2\gamma^{2H}.
$$

From (N.5) we know that

$$
\frac{C_\gamma m}{B}\leq\frac{32L_J^2}{4+\frac{\mu_F}{G^2}},
$$

which gives us

$$
\frac{1}{Sm}\sum_{s=0}^{S-1}\sum_{t=0}^{m-1}\mathbb{E}[\|w_t^{j+1}-w_{t,\star}^{j+1}\|^2]
$$

$$
\leq\left(2+\frac{G^2\mu_F+G^4}{\mu_F^2}\cdot\frac{\mu_F}{4G^2+\mu_F}\right)\frac{\frac{1}{4}\left(\frac{1}{G}\frac{\varepsilon}{3}\right)^2}{2+\frac{G^2\mu_F+G^4}{\mu_F^2}\cdot\frac{\mu_F}{4G^2+\mu_F}}
$$

$$
+2\frac{1}{\mu_F^2}\frac{C_\gamma m}{B}\frac{J^{H,\star}-J^H(\theta_0)}{L_JSm}
$$

$$
+2\frac{1}{\mu_F^2}\left(1+(\frac{8G^2}{\mu_F}+2)\frac{\mu_F}{4(4G^2+\mu_F)}\right)\frac{\sigma^2}{N}
$$

$$
+4G^2R^2\left(\frac{H+1}{1-\gamma}+\frac{\gamma}{(1-\gamma)^2}\right)^2\gamma^{2H}
$$

$$
=\frac{1}{4}\left(\frac{1}{G}\frac{\varepsilon}{3}\right)^2+2\frac{1}{\mu_F^2}\frac{C_\gamma m}{B}\frac{J^{H,\star}-J^H(\theta_0)}{L_JSm}
$$

$$
+3\frac{1}{\mu_F^2}\frac{\sigma^2}{N}+4G^2R^2\left(\frac{H+1}{1-\gamma}+\frac{\gamma}{(1-\gamma)^2}\right)^2\gamma^{2H},
$$

where we have applied (N.1) in the first equality.

Let us set

$$N \geq \frac{108 G^2 \sigma^2}{\mu_F^2 \varepsilon^2},$$

$$B \geq \frac{72 C_\gamma m}{\mu_F L_J^2 \varepsilon},$$

$$Sm \geq \frac{L_J (J^{H,\star} - J^H(\theta_0)) G^2}{\mu_F \varepsilon},$$

$$\frac{1}{4} \left( \frac{\varepsilon}{3G} \right)^2 \geq 4 G^2 R^2 \left( \frac{H+1}{1-\gamma} + \frac{\gamma}{(1-\gamma)^2} \right)^2 \gamma^{2H},$$

(N.7)

so that

$$\frac{1}{Sm} \sum_{s=0}^{S-1} \sum_{t=0}^{m-1} \mathbb{E}[\|w_t^{j+1} - w_{t,\star}^{j+1}\|^2] \leq \left( \frac{1}{G} \frac{\varepsilon}{3} \right)^2$$

By Jensen's inequality, we further have

$$\left( \frac{1}{Sm} \sum_{s=0}^{S-1} \sum_{t=0}^{m-1} \mathbb{E}[\|w_t^{j+1} - w_{t,\star}^{j+1}\|] \right)^2 \leq \frac{1}{Sm} \sum_{s=0}^{S-1} \sum_{t=0}^{m-1} \left( \mathbb{E}[\|w_t^{j+1} - w_{t,\star}^{j+1}\|] \right)^2$$

$$\leq \frac{1}{Sm} \sum_{s=0}^{S-1} \sum_{t=0}^{m-1} \mathbb{E}[\|w_t^{j+1} - w_{t,\star}^{j+1}\|^2] \qquad (N.8)$$

$$\leq \left( \frac{1}{G} \frac{\varepsilon}{3} \right)^2.$$

where we have also applied $\left( \mathbb{E}[\|w_t^{j+1} - w_{t,\star}^{j+1}\|] \right)^2 \leq \mathbb{E}[\|w_t^{j+1} - w_{t,\star}^{j+1}\|^2]$.

- Bounding $\frac{1}{Sm} \sum_{s=0}^{S-1} \sum_{t=0}^{m-1} \|w_t^{j+1}\|^2$.
  By (N.6) we have

$$\frac{1}{Sm} \sum_{s=0}^{S-1} \sum_{t=0}^{m-1} \mathbb{E}[\|w_t^{j+1}\|^2]$$

$$= \frac{1}{Sm} \frac{1}{\eta^2} \sum_{s=0}^{S-1} \sum_{t=0}^{m-1} \mathbb{E}[\|\theta_{t+1}^{j+1} - \theta_t^{j+1}\|^2]$$

$$\leq \frac{J^\star - J(\theta_0)}{L_J \eta^2 Sm} + \left( \frac{8G^2}{\mu_F} + 2 \right) \frac{\mu_F}{128 G^2 \eta^2 L_J^2} \frac{\sigma^2}{N}$$

$$+ \left( \frac{8G^2 \mu_F}{4} + \frac{8G^4}{4} \right) \frac{\mu_F}{128 \eta^2 G^2 L_J^2} \frac{1}{Sm} \sum_{s=0}^{S-1} \sum_{t=0}^{m-1} \mathbb{E}\|F^{-1}(\theta_t^{j+1}) u_t^{j+1} - w_t^{j+1}\|^2$$

$$\leq \frac{J^\star - J(\theta_0)}{L_J \eta^2 Sm} + \left( \frac{8G^2}{\mu_F} + 2 \right) \frac{\mu_F}{128 G^2 \eta^2 L_J^2} \frac{\sigma^2}{N} + \frac{\varepsilon}{9 M \eta},$$

where we have applied (N.6) in the first inequality, and (N.1) in the last step.
We can set

$$N \geq \frac{9 M \mu_F (\frac{8G^2}{\mu_F} + 2) \sigma^2}{128 G^2 \eta L_J^2 \varepsilon},$$

$$Sm \geq \frac{9M(J^\star - J(\theta_0))}{L_J \eta \varepsilon},$$

(N.9)

so that

$$\frac{1}{Sm} \sum_{s=0}^{S-1} \sum_{t=0}^{m-1} \mathbb{E}\|w_t^{j+1}\|^2 \leq \frac{\varepsilon}{3 M \eta}.$$

(N.10)

- Bounding $\frac{1}{Sm}\mathbb{E}_{s\sim d_\rho^{\pi^\star}}\left[\mathrm{KL}\left(\pi^\star(\cdot\,|\,s)||\pi_{\theta^0}(\cdot\,|\,s)\right)\right]$.

  Let us set

$$Sm \geq \frac{3\mathbb{E}_{s\sim d_\rho^{\pi^\star}}\left[\mathrm{KL}\left(\pi^\star(\cdot\,|\,s)||\pi_{\theta^0}(\cdot\,|\,s)\right)\right]}{\eta\varepsilon} \tag{N.11}$$

  so that

$$\frac{1}{\eta Sm}\mathbb{E}_{s\sim d_\rho^{\pi^\star}}\left[\mathrm{KL}\left(\pi^\star(\cdot\,|\,s)||\pi_{\theta^0}(\cdot\,|\,s)\right)\right] \leq \frac{\varepsilon}{3}. \tag{N.12}$$

By combining (N.8), (N.10), (N.12) and (J.2), we can conclude that

$$J(\pi^\star) - \frac{1}{K}\sum_{k=0}^{K-1} J(\theta^k) \leq \frac{\sqrt{\varepsilon_{\text{bias}}}}{1-\gamma} + \varepsilon.$$

To achieve this, we require $Sm$, $B$, and $N$ to satisfy (N.5), (N.7), (N.9), and (N.11), which leads to

$$Sm = \mathcal{O}\left(\frac{1}{(1-\gamma)^2\varepsilon}\right), \qquad N = \mathcal{O}\left(\frac{\sigma^2}{\varepsilon^2}\right),$$

$$B = \mathcal{O}\left(\frac{W}{(1-\gamma)\varepsilon}m\right), \qquad H = \mathcal{O}\left(\log\left(\frac{1}{(1-\gamma)\varepsilon}\right)\right).$$

By Proposition I.2, we know that in order to achieve (N.1), SGD requires sampling $\mathcal{O}\left(\frac{1}{(1-\gamma)^4\varepsilon^2}\right)$ trajectories per iteration.

Therefore, by taking $S = \mathcal{O}\left(\frac{1}{(1-\gamma)^{2.5}\varepsilon^{0.5}}\right)$ and $m = \mathcal{O}\left(\frac{(1-\gamma)^{0.5}}{\varepsilon^{0.5}}\right)$, the amount of trajectories required by SRVR-NPG is

$$S\left(N + mB + (1+m)\mathcal{O}\left(\frac{1}{(1-\gamma)^4\varepsilon^2}\right)\right)$$

$$= \mathcal{O}\left(\frac{\sigma^2}{(1-\gamma)^{2.5}\varepsilon^{2.5}} + \frac{W}{(1-\gamma)^{2.5}\varepsilon^{2.5}} + \frac{1}{(1-\gamma)^6\varepsilon^3}\right).$$

It is straightforward to verify that the requirements listed in (N.2) are also satisfied as long as $\varepsilon$ is small enough.

## O   Implementation Details

In this section, we provide additional details on the implementation of PG, NPG, SRVR-PG and SRVR-NPG.

1. For NPG, we use the default implementation provided by rllab[1], which actually implements the trust region policy optimization(TRPO) algorithm [5]. For cartplole, we sample 200 trajectories at each iteration to solve the subproblem of TRPO. For mountain car, we sample 120 trajectories at each iteration.

2. We found that the naive implementation of PG and SRVR-PG typically do not work for our tests. For example, PG and SRVR-PG often give an average reward around $-90$ for the mountain-car test, despite of our best efforts.

3. As in [19] and [21], we found that it is necessary to apply Adagrad [62] or Adam [63] type of averaging to improve their performances.

4. In our experiments, we apply Adagrad type of averaging for PG and SRVR-PG, which results in much better performances. As for SRVR-NPG, we apply Adam type of averaging, which gives an approximation of the Fisher information matrix at each iteration (see section 11.2 of [53]). We leave the implementation of a better approximation of the Fisher information matrix to the future work.