[Reviews · NeurIPS 2020]

Review 1

Summary and Contributions: This paper propose a variance-reduced natural policy gradient method, i.e., SRVR-NPG. Moreover, it studies the global convergence properties of the existing PG, NPG, SRVR-PG methods and the proposed SRVR-NPG method.

Strengths: Although the paper studies the global convergence properties of the existing PG, NPG, SRVR-PG methods and the proposed SRVR-NPG method, these theoretical analysis mainly come from the existing work (Agarwal et al., 2019). At the same time, the proposed SRVR-NPG method is also based on the exiting SRVR-PG method (Xu et al., 2020) and the exiting NPG method (Agarwal et al., 2019). In addition, this paper does not provides any experimental results to demonstrate the efficiency of the proposed methods.

Weaknesses: This paper does not provides any experimental results to demonstrate the efficiency of the proposed methods.

Correctness: The methods in this paper are basically correct.

Clarity: This paper is basically well written.

Relation to Prior Work: In fact, the theoretical analysis in the paper mainly come from the existing work (Agarwal et al., 2019). So the paper should point some differences in the proofs.

Reproducibility: No

Additional Feedback: Some comments are given as follows: C1: The proposed SRVR-NPG method uses the Fisher information. Why dose it only reach the same complexity complexity as the existing SRVR-PG without using the Fisher information ? C2: Assumption 4.3 is a strict assumption. If you must use this assumption, you should point its disadvantage. C3: To demonstrate the effectiveness of our methods, you should provides some experimental results. ------------------------------------------------------------------------------------------------------------- Thanks for your responses. I decide to increase the scoring. I also hope you should add some more related references: [1] A Hybrid Stochastic Policy Gradient Algorithm for Reinforcement Learning, AISTATS 2020, https://arxiv.org/pdf/2003.00430.pdf [2] Momentum-Based Policy Gradient Methods, ICML 2020, https://arxiv.org/pdf/2007.06680.pdf


Review 2

Summary and Contributions: The paper introduces a framework for analyzing the global convergence of PG methods and their variance-reduced variants (for state-of-the-art SRVR-PG and the paper introduces SRVR-NPG) under general assumptions. The global convergence of NPG is improved from O(eps^{-4}) to O(eps^{-3}) up to inherent errors due to policy parametrization.

Strengths: The paper seems theoretically solid and pushes state-of-the-art understanding. A framework for global convergence as mentioned above is new and was a missing piece.

Weaknesses: The paper may wish to discuss other policy estimators and how their work may apply or leave open problems.

Correctness: The results seem correct.

Clarity: The paper is well-written and necessarily contains a lot of details.

Relation to Prior Work: Related work seems clearly discussed.

Reproducibility: Yes

Additional Feedback: Line 99: Q^pi


Review 3

Summary and Contributions: The paper analyzes the global convergence of stochastic policy gradient and natural policy gradient methods, as well as their variance-reduced versions. The variance-reduced version of natural policy gradient is novel, as are the global convergence rates for the variance-reduced algorithms. Moreover, the global convergence rate for natural policy gradient improves upon existing state-of-the-art results. The global convergence rate analysis leverages the connection between the natural gradient direction and compatible function approximation as in Agarwal et al (2019), resulting in a bound composed of an approximation error term, a stationary convergence term, and a term relating the direction chosen by the algorithm to the true natural gradient. This general framework allows previous stationary and global convergence analyses to be combined.

Strengths: The analysis of global convergence in the reinforcement learning setting is very interesting given the non-convex and stochastic nature of the problem. The authors derive a general global convergence bound (Proposition 4.5) that separates the analysis into intuitive components that can be analyzed separately for a given algorithm. A theoretical understanding of the global convergence properties of policy gradient algorithms is important, as it can help to explain the observed empirical performance and provide insight into potential ways to improve existing methods.

Weaknesses: The paper’s main contributions are theoretical, but all of the theoretical analysis is placed in the supplementary material. It would be beneficial to include additional detail in the main paper, particularly on the general framework of Proposition 4.5 and the improved convergence rate of Theorem 4.8. Some of the parameter choices used in the global convergence proofs require values that are based on the unknown optimal policy, such as the KL divergence between the initial and optimal policies as well as the difference between the initial and optimal policy values. This results in implicit bounds, rather than upper bounding these values to provide explicit bounds. In addition, the dependence on the KL divergence between initial and optimal policies can lead to a vacuous bound if these policies do not share the same support (although for commonly chosen policy parameterizations, this is not an issue). While the global convergence is a nice claim, it critically depends on the policy parametrization being \epsilon_{approx} close to the optimal (Ass. 4.4). If that is large, essentially we have no global guarantees. From this perspective, the results are not too different in practice from earlier first-order guarantees (which make no assumptions on the expressiveness of the parametrization).

Correctness: There do not appear to be any obvious errors. Proofs in the supplementary material were reviewed, but not thoroughly checked for correctness.

Clarity: The paper is well-organized, and the authors clearly state the motivation for their work as well as their main contributions and results. The problem setup is explained thoroughly, and some intuition behind the structure of the theoretical contributions is given. The introduction and conclusion mention that the theoretical analysis integrates the advantages of previous work on policy gradient and natural policy gradient convergence in a way that they improve upon each other. This is a very interesting claim, and it appears that the authors are referencing the form of Proposition 4.5. However, the connection between this claim and the theoretical results is not clearly explained in the paper.

Relation to Prior Work: The authors do a very nice job of summarizing prior work on convergence rates for policy gradient methods and variance-reduction techniques. The paper’s main contributions relative to prior work are clearly discussed. The authors note that parts of their analysis closely follow the work of Agarwal et al (2019). It appears that the form of Proposition 4.5 is the novel contribution relative to this prior work, which allows for the global convergence analysis of variance-reduced methods and the improved convergence rate of natural policy gradient. Additional commentary on their contributions relative to Agarwal et al (2019) would add clarity.

Reproducibility: Yes

Additional Feedback: The paper provides a very interesting theoretical contribution on the global convergence rates of policy gradient methods in reinforcement learning. Prior work is well summarized and results are clearly stated. As mentioned previously, it would be nice to see some additional theoretical analysis in the main paper rather than deferring all proofs to the supplementary material. The extensive Preliminaries section could be shortened to provide adequate space to include this analysis. *********** Comments after the rebuttal ********** I have read the rebuttal and it reinforces my positive review.


Review 4

Summary and Contributions: The paper theoretically gives sample complexity of PG, SRVR-PG, NPG, for the global convergence, under an assumption of positive definite Fisher information matrix. A new algorithm is proposed as SRVR-NPG.

Strengths: The paper generalizes the work on global convergence of NPG to PG and SRVR-PG, and improve the result on NPG. Comparing to the previous results on the convergence to a first-order stationary point, the global convergence is stronger and is desired by practitioners. This work can be useful for the theoretical reinforcement learning community.

Weaknesses: From the theoretical point of view, is the proved sample complexity tight? Is there a way to measure the error due to the policy parameterization? The paper proposes a new algorithm SRVR-NPG, but there is no experiment to demonstrate its empirical performance. Can the proved sample complexity be validated by simulation, even in simplified cases such as the Gaussian policy? In practice, the policy gradient methods often exhibit high variance in results and sensitivity to random initializations. Is this fact empirically inconsistent with the global convergence claim made in the paper? --- Post Rebuttal comment: I will maintain the previous review.

Correctness: As far as the reviewer can tell, it is correct. But the reviewer doesn’t go over all technical details of the proofs in the appendix.

Clarity: Yes.

Relation to Prior Work: Yes.

Reproducibility: Yes

Additional Feedback:


Review 5

Summary and Contributions: This is a theoretical paper focusing on improved analysis of the sample complexity of policy gradient algorithms, including both natural gradients and variance reduction methods. POST-REBUTTAL I appreciate the authors' detailed response to the technical points raised in my review. Further (theoretical and/or empirical) analysis of the tightness of the bounds derived would strengthen the paper further, but I'm happy to raise my score to a 7 based on the rebuttal.

Strengths: The paper is clearly written, and these results ought to be of interest to the theoretical RL community at NeurIPS. I also found that the proofs in the supplementary material were generally well written and easy to follow. As far as I am aware, the analysis is novel and leads to new, strengthened results relative to prior work, although there is some overlap in the techniques employed.

Weaknesses: At present there are a few technical aspects of the paper that I am unsure about, and so would appreciate clarification from the authors on these; they are described in detail below. My current recommendation for the paper reflects this uncertainty, and I will update it after seeing the authors' response. Beyond this, I felt that although the paper does a good job of citing earlier related works and associated results, the paper would be improved with more discussion of the assumptions made in this paper, how these compare to earlier works, and also with more discussion of how the results stated in these earlier works can be translated into the form cited in e.g. Table 1. Finally, it would have been interesting to see some empirical work to complement the theoretical results here. I don't think this should be necessary for acceptance, and I think perhaps the most interesting experiments here would not be large-scale application of these algorithms, but simple experiments on small environments to understand how tight the sample complexity bounds are, and in particular whether the SRVR-NPG analysis is likely to be sub-optimal (since it matches the complexity of NPG in Table 1).

Correctness: I have reviewed the proofs of Proposition 4.5 and Theorem 4.6 (as well as the results in the appendix they depend on) line-by-line, and believe these to be broadly correct, although there are some suspected typos I have listed below in more detail, including in the theorem statement. I also had a query about the choice of H in the proof which I think may require the statement to modified slightly. I was not able to review more proofs at this level of detail due to time constraints and the length of the paper, however I have made high-level checks of the remaining results in the main paper. The assumptions made in the theoretical results generally seem reasonable (relative to necessary assumptions in other work), although I have a few specific comments on these below that I would appreciate clarification on from the authors. I will revise my recommendation after seeing the authors' rebuttal in response to these points.

Clarity: Overall I found the paper to be written clearly. There were several places where I thought more discussion was warranted. A few examples are the sample complexities from prior works in Table 1. These take a bit of work to derive from the cited papers, so an appendix section showing how they are derived would be useful. I would also like to have seen more comparison of the assumptions made in this work with assumptions in cited works; for example, Assumption 2.1 and in particular comparison of this assumption to e.g. Assumption 6.2 in [1].

Relation to Prior Work: The authors do a good job of putting their work in context against earlier work on policy gradient methods, and variance reduction methods for optimisation more generally.

Reproducibility: Yes

Additional Feedback: Detailed comments on a section-by-section basis are given below. Lines 10-13 of abstract: I found this quite vaguely worded, and didn't understand which contributions in the main paper this is referring to. Section 1 "Policy gradients are usually estimated via Monte-Carlo rollouts". To me, this would suggest that critics are not used, or are used at most as baselines. However, as far as I am aware, most of the successful methods cited in the previous paragraph do use bootstrapping, and so are not pure Monte Carlo, in the sense that the term is used in RL. Table 1 caption: "the expected number of trajectories to reach eps-optimality". Should this be "the number of trajectories to reach an eps-optimal policy in expectation"? Section 2 Eqn (2.10): clarify that the squaring happens inside the expectation - it is a little ambiguous at the moment, and actually reads more like the expectation itself is squared. Section 3 Similarly to Eqn (2.10), can the authors clarify that the squaring that appears in the objective of Eqn (3.3) should be inside the expectation? Section 4 Assumption 4.1: This could perhaps be stated a little more precisely: "variance" is a little bit ambiguous for a vector, and what is presumably meant is the expectation of the L^2 norm of the RV minus its mean? Assumption 4.3: This seems like a very strong condition, and it seems as though this will not be satisfied by the family of Gaussian policies, nor for softmax parametrisations of the simplex. Proposition 4.5 proof. Why is the final line of (I.1) preceded by an inequality? Doesn't w^k_* attain the minimum of the of compatible function approximation error, meaning that we can have an equality here? Theorem 4.6 The quantity L_J is not defined in the main paper as far as I can tell, but in Appendix A. Please consider moving the definition to the main paper, or at least adding a pointer to the section of the appendix where it is defined. If we require K = O((1-gamma)^{-4} eps^-2}) and N = O(sigma^2 (1-gamma)^{-2} eps^{-2}), why isn't the overall number of trajectories equal to O(sigma^2 (1-gamma)^{-6} eps^{-4})? Judging from the proof in the appendix, I expect there are some typos in the theorem statement in the main paper as to these trajectory complexities - please can the authors clarify? Remark 4.9: As far as I can tell, the result in [1] also uses constant stepsizes, which doesn't seem to align with the comment here; can the authors comment on this? Lemma A.1: In the penultimate calculation line, the expression GR \sum_{h=H}^\infty h \gamma^h shouldn't have \| \| around it. I think there is a missing factor of \gamma in the first term of the bound GR(H/(1-\gamma) + \gamma/(1-\gamma)^2)\gamma^H, that should multiply the "H". This doesn't affect the validity of the upper-bound derived in this lemma, and so should not affect the validity of the analysis that depends on Lemma A.1, but it is not as tight as it could be. Theorem 4.6 proof I think there is an additional factor of 2 in the final line of (J.4) which is not required (since 2 premultiplies (1+1/\mu_F)^2, but also appears in the next bracket. I think this only makes the bound looser, and doesn't make it invalid,but clarification on this would be appreciated. Line 685: L_j -> L_J. In (J.5), it is not clear to me that such a choice of H is possible, since the right-hand side is bounded above as a function of H. In particular, when G is small relative to epsilon, there may exist no valid choice of H. This may be a misunderstanding on my part, so could the authors clarify the choice of H here, and whether some condition on epsilon being sufficiently small relative to other parameters such as G is required? References: [1] has recently been updated on arxiv (with a new title, and rearranged content). Can the authors ensure that future versions of their paper refers to the new version of [1], or perhaps include the arxiv version number in the reference if the authors want to continue to refer to the currently cited version.

[Author Response · NeurIPS 2020]

R1.1...*these analysis mainly come from the existing work...the novelty is very limited.* We respectfully disagree.
As pointed out by R2, R3, R4, and R5, this paper develops a novel global convergence *framework* that *unifies* the
convergence of several policy gradient methods, whose novelty is summarized in Lines 210-212 and further explained
in Lines 216-225. We proved the *global convergence* of SRVR-PG, for the first time; and improved the $\mathcal{O}(\varepsilon^{-4})$ sample
complexity of NPG into $\mathcal{O}(\varepsilon^{-3})$. Our proposed SRVR-NPG has a better complexity than SRVR-PG (Remark 4.13).

Cartpole performance averaged over 10 runs, NN size = 32

R1.2 *...experimental results...* This paper focuses on laying the theoretical foundation for the global convergence of policy gradient methods, as [1,15,26,47]. Note that none of [1,15,26,47] has numerical results. We believed our theoretical contrition already has archival value. But still, we include a numerical result in the figure on the left. As this rebuttal will be archived in the NeurIPS proceeding, we assure this and more numerical results will be added, as well as more simulation details.

7
R1.3 *Reproducibility:* We believe that all of our theoretical claims have been proved. R1.4 *SRVR-NPG same complexity*
*as SRVR-PG?* SRVR-NPG has a better iteration complexity, but it needs to sample trajectories to solve a subproblem
at each iteration. Overall, its complexity has a better dependence on $W$ and $\sigma^2$. R1.5 *Assumption 4.3 is a strict*
*assumption...* As pointed out in Lines 199-201, this assumption is standard in the analysis of variance-reduced policy
gradient methods, and it can be verified for Gaussian policies. Please refer to [34] for a detailed proof.

R1 As the main concerns are regarding the novelty and numerical validations, with our clarifications and new simulation
results, we appreciate that the reviewer would re-evaluate our contribution, and change the scoring accordingly.

R2.1 *...other policy estimators...* Will add more discussions on this and open problems, when an extra page is available!

R3.1 *...It would be beneficial to include additional detail in the main paper...* Thanks for the suggestion. Proposition 4.5
applies the performance difference lemma and connects the global convergence rate with the stationary convergence
rate. We will add more explanations in addition to Lines 210-212 and 240-243. R3.2 *...if these policies do not share the*
*same support...* We agree that they should share the same support. Will add this. R3.3 *the global convergence...critically*
*depends on $\varepsilon_{approx}$...* The richness of the function class explicitly occurs in the error bound, which will become very
small or even zero under many common parametrizations, e.g., softmax, overparametrized neural nets, etc. This global
convergence of RL cannot be provided by first-order guarantees, so it is much stronger than the latter in this sense.

R4.1 *...is the proved sample complexity tight?...the error due to the policy parametrization?* Interesting question. To the
best of our knowledge, there isn't any lower bound for policy gradient methods under general policy parametrizations
yet. We use $\varepsilon_{\text{approx}}$ to characterize the error due to policy parametrization (see Lines 204-208). R4.2 *...no experiment to*
*demonstrate its empirical performance...* Please refer to R1.2 for some numerical results. R4.3 *...policy gradient methods*
*exhibit high variance...inconsistent with the global convergence...?* Our results require sampling more trajectories per
iteration than what's typically done in practice (e.g., $\mathcal{O}(\varepsilon^{-2})$ for PG). This will *stabilize* the performance, so we believe
there is no inconsistency/counter-intuition from practice.

R5 Your detailed and thoughtful review is very helpful for us! Hope that our response will address your concerns.

R5.1 *...more discussion of the assumptions...how the results stated in earlier works can be translated...* We will make the
presentation better, and make the translations more explicit. R5.2 *it would have been interesting to see some empirical*
*work...whether the SRVR-NPG analysis is sub-optimal.* We have some numerical results in R1.2. Will add more to
see if the analysis is tight or not. R5.3 *...more discussion...compare Assumption 2.1 with Assumption 6.2 in [1].* Will
add more discussions. Specifically, the Assump 6.2 in the [1] (updated recently) implies our Assump 2.1. We found
this independent but related finding quite interesting, and will discuss this. R5.4 *Section 2,3*: We have corrected the
typos. R5.5 *...usually estimated via Monte-Carlo...* Sorry for the confusion. We will change the wording here. R5.6
*Assumption 4.1* In stochastic optimization, "variance" refers to the expectation of $L^2$ norm of the bias. Will clarify.
R5.7 *Assumption 4.3* As mentioned in Lines 199-201, this assumption is standard in the analysis of variance-reduced
policy gradient methods [34, 51, 52], and can be verified for Gaussian policies. Please refer to [34] for a detailed proof.
R5.8 *Proposition 4.5* In Assumption 4.4, $\varepsilon_{\text{approx}}$ is an upper bound of *all* compatible function approximation error. R5.9
*Theorem 4.6* $K$ should be $O((1-\gamma)^{-2}\varepsilon^{-2})$ and $N$ should be $O(\sigma^2\varepsilon^{-2})$. Sorry for the confusion! Will define $L_J$.
R5.10 *Remark 4.9* Yes, [1] does apply a small constant stepsize $\eta = \mathcal{O}(T^{-0.5})$, but with $T = \mathcal{O}(\varepsilon^{-2})$, not an absolute
constant as ours. R5.11 *Lemma A.1* We believe that the calculation $\sum_{h=H}^{\infty} h\gamma^h = \left(H/(1-\gamma) + \gamma/(1-\gamma)^2\right)\gamma^H$ is
correct. R5.12 *...an additional factor of 2...* There is a factor of 2 in the final line of (J.1) and (J.3), so in total we need 4.
R5.13 *In (J.5),...choice of H...* We apologize for the confusion. The $\leq$ in the first equation of (J.5) should be $\geq$, it is a
typo. We choose $H = \mathcal{O}\left(\log((1-\gamma)^{-1}\varepsilon^{-1})\right)$ so that the right-hand side is upper bounded by $\frac{1}{3}\left(\frac{\varepsilon}{3G}\right)^2$. R5.14 *...the*
*new version of [1]...* Thanks for the notification. We will update our paper. Remarkably, with the new Assumption 6.5
of [1], we can simplify the analysis, and analyze the original NPG update without resorting to $\tilde{J}(\theta)$.

[Meta-Review · NeurIPS 2020]

The results in this paper were deemed interesting. The reviewers were left somewhat uncertain, because it wasn't always clear exactly how significant these results will turn out to be (how limiting are the assumptions, how does this pan out in practice), but were still okay with accepting the paper as is, thus allowing the research community to build further to help answer some of these questions in the future.